# Validation of automatic monitoring of feeding behaviours in sheep and goats

**Roxanne Berthel** [1]*, **Alisha Deichelboher**[1], **Frigga Dohme-Meier**[2], **Wendelin Egli**[3], **Nina Keil**[1]

**1** Centre for Proper Housing of Ruminants and Pigs, Federal Food Safety and Veterinary Office, Agroscope, Ettenhausen, Switzerland, **2** Ruminant Research Group, Agroscope, Posieux, Switzerland, **3** MSR Electronics GmbH, Seuzach, Switzerland

* roxanne.berthel@agroscope.admin.ch

## Abstract

Monitoring the feeding and ruminating behaviour of ruminants can be used to assess their health and welfare. The MSR-jaw movement recording system (JAM-R) can automatically record the jaw movements of ruminants. The associated software Viewer2 was developed to classify these recordings in adult cattle and calculate the duration and number of mastications of feeding and ruminating. The purpose of this study was to evaluate the performance of Viewer2 in classifying the behaviour of sheep and goats and assessing their feeding and ruminating. The feeding and ruminating behaviour of ten sheep and ten goats on pasture (observed live) and of five sheep and five goats in the barn (observed by video) were compared with Viewer2 behaviour classifications. To assess the technical and welfare issues of the JAM-R, its application was tested in a feeding experiment with 24 h monitoring of the feeding behaviours of 24 sheep and 24 goats. Viewer2 worked equally well on both species. The mean (95% confidence interval) performance of Viewer2 was at a good level for feeding (accuracy: 0.8–1.0; sensitivity: 0.9–1.0; specificity: 0.6–0.9; precision: 0.7–0.9) and ruminating (accuracy: 0.8–0.9; sensitivity: 0.6–0.8; specificity: 0.8–1.0; precision: 0.9–1.0) compared with human observations, with minor differences between the conditions on pasture and in the barn. The performance improved when recording frequency was increased from 10 Hz to 20 Hz. Applying the JAM-R in a feeding experiment, 71% of the recordings executed were defined as technically error-free and produced plausible values for feeding behaviours. In conclusion, according to the values of accuracy, sensitivity, specificity and precision, the presented JAM-R system with Viewer2 is a reliable and applicable technology for automatic recording of feeding and ruminating behaviour of sheep and goats on pasture and in the barn.

## 1 Introduction

Good animal health and welfare are prerequisites for high-quality research and efficient animal-friendly commercial livestock production [1]. In ruminants, monitoring feeding behaviours is useful for assessing their health and welfare. Different measures of feeding behaviours,

**Data Availability Statement:** All relevant data are within the paper and its Supporting Information files.

**Funding:** The project was funded by the Swiss Federal Food Safety and Veterinary Office (project

number 2.19.e). The funders had no role in study design, data collection and analysis, decision to publish, or preparation of the manuscript.

**Competing interests:** The authors have declared that no competing interests exist.

such as feeding (actual feed intake) and ruminating (regurgitation, chewing and swallowing of a cud of ingested food), can be used as indicators. For instance, the duration of feeding and ruminating or the number of mastications made are used to evaluate feed qualities (e.g. fibre content [2] or forage quality [3]]. They can also indicate the animals' health status [4] or rumen health risks [5]. Furthermore, inter-animal variation in feeding duration and meal frequency can indicate social stress or feed competition in the herd, which may be caused by feed delivery frequency [6] or dominance hierarchies [7]. The detection of a change in these indicators requires continuous long-term recording (for a few hours up to several days) of behaviour at the individual or herd level. Despite the rise of technological solutions, behaviour observation by human observers is still commonly used to record animal behaviour and can be considered the "gold standard". However, it requires training and is very time consuming, making it challenging in research and very demanding on farms.

Automated recording methods can substitute the human observer provided they are equally good at detecting feeding behaviour as humans. In the past, automatic monitoring systems were developed based on a variety of mechanisms, like head position with accelerometers or jaw movements with acoustic or pressure sensors [8]. Devices that record actual jaw movements with pressure-sensitive sensors have been designed repeatedly because this approach has a number of advantages [9]. First, it measures behaviour directly at the individual level, so the data can always be linked to the individual, independent of the animal's location (barn or pasture). Second, the available sensors are small enough to fit into common head collars, so animals can easily adapt to the system. Third, the measurement of jaw movements is at the exact location where the behaviour of interest originates. Therefore, it is even possible to record single bites or chews (mastications). This enables a distinction between feeding and ruminating, as the duration and frequency of mastication differ between these two behaviours [10,11].

Based on jaw movements, the MSR-jaw-movement recording system (JAM-R) was developed by the Swiss research centre Agroscope and MSR Electronics GmbH (Seuzach, Switzerland) to record the feeding behaviour of adult cattle [12]. The JAM-R technology has previously been tested and validated for dairy cows [13]. Later, the Viewer2 software (freely available on request from MSR Electronics GmbH, Seuzach, Switzerland) was developed to automatically classify feeding and rumination behaviour and calculate the duration and number of mastications (defined as all kinds of bites and chews) based on the R analysis described by Nydegger et al. [12].

In many parts of the world, sheep and goats constitute a major proportion of agriculturally used livestock [14]. The same line of reasoning for monitoring feeding and ruminating behaviour in cattle applies to these species. Some automatic behaviour recording methods have already been successfully transferred from cattle to sheep and goats [15,16], even though certain feeding parameters differ between these species. For instance, differences were found in browsing behaviour [17,18], bite size and rate [19], and the number of mastications per ingested feed unit [20]. In a research project, the halter of the JAM-R was adapted to the size of goats, and the recorded data could be successfully used to monitor feeding and ruminating [21]. However, whether the logarithm of Viewer2, which was developed to calculate the feeding and ruminating of cattle, would perform equally well for sheep and goats was not validated. Additionally, the calculations of Viewer2 were validated with a recording frequency of 10 Hz [13], while the JAM-R offers a choice of recording frequencies up to 50 Hz. As sheep and goats make more mastications per minute than cattle [20], increasing recording frequency could improve the performance of Viewer2 for these species.

The purpose of this study was to evaluate the performance of the JAM-R in calculating the feeding and ruminating behaviour of sheep and goats. The study investigated whether Viewer2

was able to correctly classify these behaviours on pasture and in the barn, and whether the recording frequency (10 Hz or 20 Hz) had an effect on classification performance. Additionally, the system was applied to sheep and goats in a feeding experiment on three different mixed rations (MRs) to assess the plausibility of obtained values and report frequencies of technical failure and possible problems for the welfare of the animals due to the wearing of the halter.

## 2 Materials and methods

To validate the JAM-R, the behaviour classifications of Viewer2 software (described in Section 2.3.2) were compared to the gold standard method of behaviour recording, which is the observation by a human (detailed methods used is described in Section 2.2). The feeding and ruminating behaviour of goats and sheep was recorded by direct live observations on pasture and video observations in the barn in experimental pens, while the animals were equipped with the JAM-R. The agreement between the classifications of Viewer2 and the behaviour observations was evaluated. Additionally, JAM-R was applied in a feeding experiment with multiple 24 h recordings and assessed for technical and welfare issues (Section 2.5).

All animal care and experimental procedures were performed in accordance with the relevant legislative and regulatory requirements and the ASAB/ABS Guidelines for the Use of Animals in Research [22]. The Cantonal Veterinary Office, Thurgau, Switzerland (Approval No. TG10/18–30902) approved all procedures involving animal handling and treatment.

### 2.1 Animals and housing

**2.1.1 Non-experimental condition.**   All experimental animals were part of two stationary herds of 26 sheep and 27 goats at the Agroscope Tänikon Research Centre, Ettenhausen, Switzerland. The sheep were of the dairy breeds Lacaune (LC) and East Friesian (EF). The goats were of the dairy breeds Saanen (SA), Chamois Coloured (CC) and hybrids of these two breeds (HY). All animals were around three years of age, non-lactating and not pregnant. The mean weight of goats was 67.7 (SD ± 7.5) kg and of sheep was 79.1 (± 7.7). The experimental animals were accustomed to human handling from previous experiments and had been feeding on pasture and different types of roughage (hay, grass silage, maize silage) before the experiment.

Between the experimental phases and during the feed habituation phases, the animals were housed in one group of each species and fed on roughage ad libitum. The home pen for the goats had a total area of 53 m$^2$ (13.6 m x 3.9 m), with a 40 m$^2$ straw-bedded deep litter area and 0.95 m of elevated feeding area along the long axis of the pen. The deep litter area of the goats was equipped with three benches (2.4 m x 0.62 m, 0.6 m height) and three round tables (1.1 m diameter, 0.8 m height). The home pen of the sheep had a total area of 42 m$^2$ (11.7 m x 3.6 m), with 33 m$^2$ of deep litter and 0.8 m of elevated feeding area along the long axis of the pen. Both pens had three drinking water sources and one mineral supply site. Feed troughs with a neck rail were along the total length of the long axis of each pen.

**2.1.2 Pasture.**   The data collection for the validation on pasture was conducted with ten animals of each species (LA = 9, EF = 1, CC = 4, SA = 4, HY = 2) in September 2020. These animals were chosen according to ease of handling, assuming that they would be the least affected in their behaviour by the presence of the human observer. Unfortunately, this was not the case for one goat, so the data were obtained from only nine goats on pasture. The pasture was a side strip between a field and stream, containing shrubbery of a natural meadow with a sward height of about 30 cm. At all times, the animals had access to a hut (1.55 m height x 1.95 m width x 2.8 m length) and drinking water in a trough that was checked and refilled daily.

The week before the observation began, the goats were habituated to pasture on a 10 x 20 m plot for seven days, with the number of hours on pasture increasing from 5 h on the first day to 12 h on the fifth day. Thereafter, they stayed on pasture for 24 h. The sheep were habituated in the same way as the goats on a neighbouring 7 m x 30 m plot. During this pasture habituation phase, no observation or data collection took place. For the observation period, the habituation plot of the goats was used and it was subsequently extended to provide sufficient fresh grass to ensure normal feeding behaviour. For the observations of the goats on five consecutive days, the plot was extended by 10 m x 30 m (total plot = 10 m x 50 m). When the observation period of the goats was finished, the sheep were then moved to this plot, and it was extended by an additional 10 m x 10 m (total plot = 10 m x 60 m). On this plot, the behaviour observation of sheep was conducted for six consecutive days.

**2.1.3 Experimental pens.**   Data collection for validation in the barn was conducted from March to April 2020. During the feeding experiment, which will be further explained in Section 2.1.4., videos were recorded. Videos of five sheep and five goats (LA = 4, EF = 1, CC = 2, SA = 3) were chosen based on the availability of sufficient video image quality (e.g. lighting, animal in sight) for continuous behaviour observation. This included two goats that were also used in the pasture situation and three other goats and five additional sheep. Videos were recorded from 11:00 to 14:00 on two days for each animal.

They were housed in pairs of their species in experimental pens, each 2.4 m x 2.9 m in size. The pen partner animals could be, but were not necessarily, part of the video observation. The experimental pens consisted of a lying area bedded with sawdust and an elevated feeding area with two feeding places per pen (0.80 m feeding space per animal), separated by a solid wooden wall. The feeding fence consisted of wooden palisades. The goat pens were additionally equipped with a round table (1.1 m diameter, 0.8 m height). The animals received a MR of 55% grass silage, 40% maize silage and 5% alfalfa hay on a dry matter basis (MG) ad libitum, delivered in three portions at 09:00, 11:00 and 16:00.

**2.1.4 Feeding experiment for application of JAM-R.**   The JAM-R was applied in a feeding experiment (October 2019–April 2020) of sheep and goats when feeding on MRs to assess issues in application of the system. The experimental units of this study were 24 dairy goats and 24 dairy sheep in experimental pair pens (described above). Two sheep and two goats were exchanged for other animals during the course of the experiment.

The animals received three different MRs consisting of roughage consecutively. The first MR consisted of first- and second-cut grass hay (HH; 50:50 dry matter (DM) ratio), the second of grass silage and grass hay (GH; 50:50 DM ratio), and the third of maize silage, grass silage and alfalfa hay (MG; 40:55:5 DM ratio). More details on the MRs are provided in S1 Table 1 in S1 Table, and methods of feed analysis in S1 File. The animals were habituated to each MR in the home pens for 14 days. For the experimental phases, the animals were kept in the experimental pens for 10 days. During the experimental phase of (the last MR) MG, videos of behaviour observations (Section 2.2) were recorded.

## 2.2 Behaviour recording by observation

Feeding, ruminating, drinking and other oral behaviour (see Ethogram, Table 1) were continuously assessed equally for both species. During the behaviour observations, which is considered the gold standard, all animals were equipped with the reference behaviour recording method–the JAM-R.

On pasture, ten goats and ten sheep were directly observed for 3–5 hours per day between 08:00 and 19:00. The intention was to observe the animals for 2.5 h in the morning and 2.5 h in the afternoon, but depending on the weather conditions, the schedule was sometimes

**Table 1. Ethogram of sheep and goat behaviours under direct observation and video observation.**

| Behaviour | Description of live observation | Description of video observation |
|---|---|---|
| **Feeding** | Standing posture with the head on the ground | Standing posture with the head in the trough, and after removing the head from the trough, when chewing until animal stopped chewing for a minimum of 3 sec |
| **Ruminating** | Standing or lying with regular chewing movements interrupted by visible swallowing and regurgitation of cud with unmoving head and mouth | Standing or lying with regular chewing movements interrupted by visible swallowing and regurgitation of cud with unmoving head and mouth |
| **Drinking** | Head in the water trough | Position of the head and mouth in the direction of the drinker at a distance of the length of the animal's head to the water surface (which was not visible from the camera angle), with an unmoving head for a minimum of 1 sec |
| **Other oral activity** | Any oral movement directed at itself, other animals or objects (e.g. licking, nibbling, self-grooming) | Any oral movement directed at itself, other animals or objects (e.g. licking, nibbling, self-grooming) |

adapted to times when the temperature was cooler, as the animals were less active in the heat. The observation programme BORIS [29] was used for the direct behaviour coding. Using focal animal sampling, the observer continuously observed the animals by switching between animals in 10 min intervals or when the head of the focal animal moved out of sight. In total, 22.8 hours of behaviour of ten sheep and 22.7 hours of behaviour of ten goats, with a mean of 2.3 (± 0.17) hours per animal, were analysed.

In the barn, four cameras (CCD Ever Focus 430) were used for the video observations in the experimental pens. Each camera frame recorded two pens, for a maximum of four animals per camera. The behaviour of five sheep and five goats was recorded continuously for three hours from 11:00 to 14:00 on two days (6 h per animal), as this period was known to include sufficient events of feeding and ruminating. The Interact programme [30] was used for video behaviour coding.

The behaviour observations on pasture (direct) and in the barn (by video) were conducted by one trained observer (A.D.). Two of the videos were coded again by another trained observer (R.B.) and each second (21,600 s) was assessed for agreement. R.B. reached an agreement of 97.6–98.2% for seconds observed as feeding by A.D. and 98.5–99.9% for seconds of rumination of A.D.

## 2.3 Automatic behaviour recording

**2.3.1 MSR-jaw-movement recording system (JAM-R).** The JAM-R consists of an MSR145 data logger (MSR Electronics GmbH, Seuzach) with a pressure sensor connected to a vegetable oil-filled silicon tube (Fig 1). The system was integrated into a modified commercial head collar of a size that fit sheep and goats. The silicon tube was placed in a long fabric pocket sewn onto the noseband of the halter. The data logger was located inside a leather pocket sewn on the side of the halter (Fig 1). The connection cable between the tube and the data logger was covered with duct tape. Bending and pressing on the silicon tube caused by movements of the jaws results in a change in pressure, which is recorded at a set frequency of time by the data logger.

The data loggers had to be programmed with the MSR software (freely available on request from MSR Electronics GmbH, Seuzach, Switzerland) to choose the recording frequency of the pressure sensor (in our case, between 10 and 20 Hz) and the start and end times of the recording period. Another option (not used in this study) is the additional recording of the

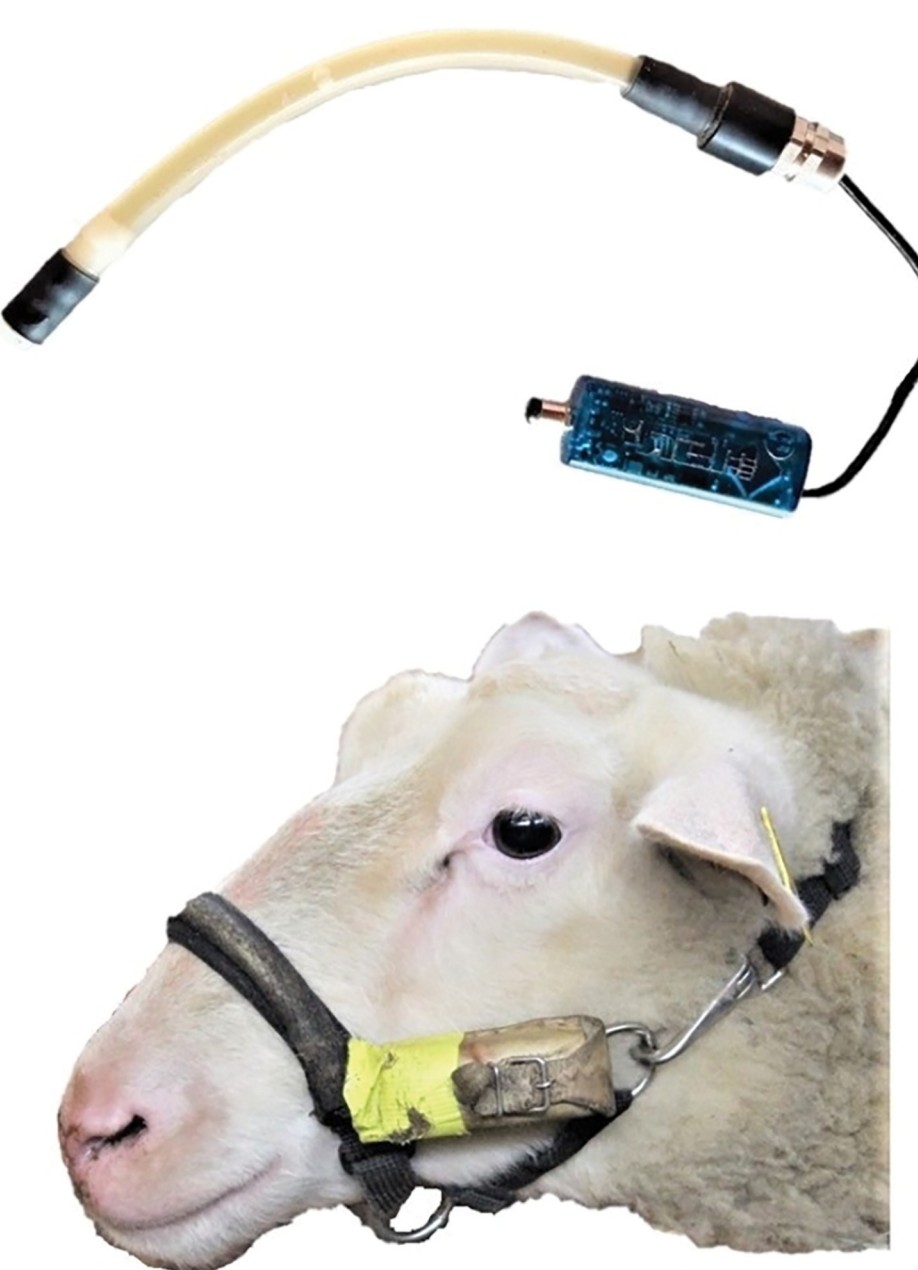

**Fig 1. MSR-jaw-movement recording system outside the halter (top) and integrated in the halter worn by a sheep (bottom).**

acceleration of one to three axes to measure the activity at freely selectable recording frequencies. The data loggers (in the version for this project) had to be removed from the animals to save the recorded data (as an MSR file) and reprogrammed for the next recording period once per day due to the data storage capacities of the loggers being a maximum of 29 hours. However, newer versions of the data loggers are available with larger data storage capacities for longer continuous recording periods from MSR electronics GmbH.

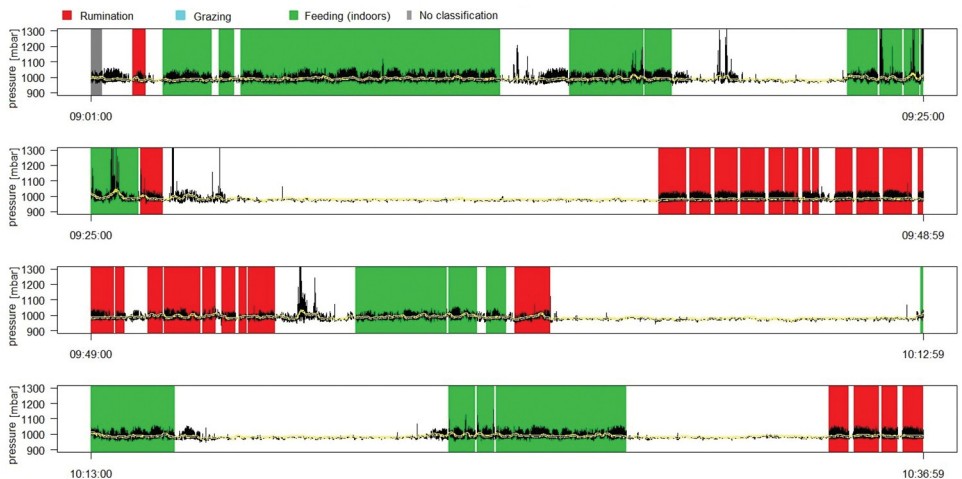

**Fig 2. Screenshot of the graphical Viewer2 output.** The pressure (mbar) is plotted against the clock time (x-axis) in a black line. The colour shaded areas indicate the periods classified as behaviour with the legend on the top. Sequences at the start (or end) of a recording that are recognised as behaviour cannot be classified ("no classification"), since the start (or end) of the behaviour is missing. The sequences classified as neither of the two behaviours are shown in white. The sequences classified as "grazing" when both behaviours occur during one recording are not represented in this file.

**2.3.2 Viewer2 behaviour classification software.** After the recording is finished, the single MSR files must be uploaded to Viewer2 to run the behaviour classifications and calculate durations and mastications. After uploading an MSR file, a characteristic feeding and ruminating sequence needs to be manually selected in Viewer2. (In case the recorded period contains times for both feeding on pasture and in the barn–which was not the case in this study–two respective feeding sequences must be selected.) Thereafter, the software automatically analyses the entire MSR file for these characteristic patterns in pressure differences and classifies each second of the JAM-R recording into "feeding", "ruminating" or "no/other behaviour". The category "no/other behaviour" includes all observed behaviours of drinking, other oral behaviours (e.g. licking, nibbling, self-grooming) and no activity. A summary output as a data file (text file format) provides the duration of ruminating and feeding (in minutes), the number of mastications for ruminating and feeding, and the number of cuds during ruminating, which can then be used to calculate the number of mastications per cud. The detailed output is provided in an Excel file, with the timestamp of every recording (e.g. every $10^{th}$ or $20^{th}$ of a second), the pressure in mbar, a logic variable for peak detection and the behaviour classification. Additionally, a graph of the pressure (in mbar), plotted against the clock time with coloured indications of ruminating and feeding sequences, was generated (Fig 2).

## 2.4 Comparison of the behaviour recording methods

The behaviour recorded by the human observer (directly on pasture and from video in the barn, see Ethogram in Tab. 1) was compared to the behaviour recorded at the same time with the data loggers and classified by the Viewer2. The data loggers were set to a recording frequency of 10 Hz when the animals were on pasture. In the barn, the recording frequency was alternated between 10 and 20 Hz, so every animal had one day of recording at 10 Hz and one day of recording at 20 Hz.

**2.4.1 Data management.** As the coding for direct and video observations with Boris and Interact is accurate to the second, the Viewer2 Excel output files were also reduced to one recording per second. From each of the ten (10 Hz) or twenty (20 Hz) recordings within 1 sec, the 10th/20th value was selected.

For the comparison of the recording methods, the agreement of every second of the Viewer2 behaviour classification with the behaviour observation at the same time point was assessed. This resulted in numbers of correctly and incorrectly classified seconds, taking the behaviour observed by the human as the true value ("gold standard"). Incorrectly classified seconds were either *false positive (FP)* when the software classified the second as a behaviour, but no behaviour (or behaviour other than feeding or ruminating) was observed, or *false negative (FN)* when the software did not classify the second as a behaviour, but a behaviour was observed. Correctly classified seconds could also be *true positive (TP)* when the classified behaviour was indeed observed or *true negative (TN)* when the software classified the second as no behaviour, and none (or behaviour other than feeding or ruminating) was indeed observed (Table 2). These numbers were calculated in error matrices using the error matrix function of the klaR package [23], the behaviour observation data as the reference and the behaviour classification data of Viewer2 as the method to be evaluated. The exact descriptions of positives and negatives of both behaviours (feeding and ruminating) and "no oral behaviour", as well as "other oral behaviour" and drinking, are added in the supplementary section (S1 Table 2 in S1 Table). The matrices were calculated for each individual and condition (pasture and barn) and frequency (10 Hz and 20 Hz). Consequently, on pasture, one error matrix was calculated per individual by summing the data of the five to six days of observations.

**2.4.2 Parameters of agreement.** With these error matrices of correctly and incorrectly classified seconds, the four parameters of agreement were calculated for each error matrix of feeding and ruminating. Accuracy (Formula a) is the proportion of correct classifications of all seconds that were analysed. It is a measurement of overall agreement. Sensitivity (Formula b) is the proportion of correctly classified behaviour from when the behaviour was observed (occurred). It is a measurement of how well Viewer2 recognised an occurring behaviour. Specificity (Formula c) is the proportion of correctly non-classified seconds when no behaviour was observed (occurred). It is a measurement of how well Viewer2 ignored "noise" in the recording and did not classify any pressure changes caused by behaviour other than feeding or ruminating. Precision (Formula d) is the proportion of correct classifications of all seconds that were classified as the behaviour. It is a measurement of how well Viewer2 distinguished the pattern of the behaviour from other pressure changes in the recording.

a) $accuracy = \frac{TP+TN}{TP+TN+FP+FN}$

b) $sensitivity = \frac{TP}{TP+FN}$

c) $specificity = \frac{TN}{TN+FP}$

d) $precision = \frac{TP}{TP+FP}$

**2.4.3 Effects of species, condition and recording frequency on agreement parameters.** With the complete dataset, in linear mixed effects models (*lmer* of lme4 package, [24]; one per parameter), as outcome variables, each parameter of agreement was estimated in dependence of the fixed effects species (sheep and goat) and behaviour (feeding and ruminating). The models also considered the repeated measurements of the individual, the effect of the condition and the recording frequency by random effects.

**Table 2. Definition of the positive and negative classifications used to assess the agreement of Viewer2 with the continuous direct and video observation recordings as reference.**

|  | Recorded behaviour in observation | |
| --- | --- | --- |
| **Classified behaviour in Viewer2** | **Behaviour present** | **Behaviour absent** |
| **Behaviour classified** | True positive (TP) | False positive (FP) |
| **No behaviour classified** | False negative (FN) | True negative (TN) |

- Parameter of agreement ~ Behaviour + Species + (1|Individual) + (1|Condition) + (1|Frequency), data = all (N = 76)

Thereafter, in two suitable subsets, analysis was conducted to determine whether the four parameters of agreement were influenced by the condition (pasture or barn) or the recording frequency (10 Hz or 20 Hz). First, by excluding the data recordings of 20 Hz, it was determined whether the respective agreement parameter (outcome variable) was affected by the condition, the behaviour or the interactions between these two factors (fixed effects), correcting for repeated measurements of the individual and the species (random effects).

- Parameter of agreement ~ Behaviour + Condition + Behaviour: Condition+ (1|Individual) + (1|Species),data = only recordings of 10 Hz (N = 56)

Second, excluding the data recordings on pasture, the influence of the recoding frequency, the behaviour and their interaction (fixed effects) on the respective parameter of agreement (outcome variable) was analysed, correcting for repeated measurements of the individual and the species (random effects).

- Parameter of agreement ~ Behaviour + Frequency + Behaviour: Frequency+ (1|Individual) + (1|Species),data = only recordings from the barn (N = 38)

The models were stepwise reduced by the interaction and then by each fixed effect and compared with the full model. This was done using the *anova* function [25], which performs a $\chi^2$ test based on the restricted maximum likelihood method to determine whether the fixed effect omitted in the reduced model is statistically significant for the model fit. The significance level of the p-value was set to 0.05.

## 2.5 Application of JAM-R in a feeding experiment

To assess issues of the applicability of the JAM-R, it was tested in a feeding experiment. Issues for animal welfare due to wearing the halters were recorded. Data loss events were reported which were caused by technical failure, animal welfare issues or human error. Raw data of the feeding and rumination durations were inspected for plausibility. The variation in the data that was caused by this set of data loggers was assessed.

**2.5.1 Description of the feeding experiment.** For the feeding experiment, each MR was fed ad libitum for ten days and offered at 9:00, 11:00 and 16:00. The amounts fed and left over the following day were weighed to calculate feed intake per pair and divided by two to estimate individual intake.

The feeding and ruminating behaviour of each animal was recorded with the JAM-R system on experimental days 4, 5, 9 and 10. The recording frequency was set to 10 Hz, except for 47 cases in which it was set to 20 Hz for the validation described in Section 2.4, which were recorded during the last MR of this experiment. The recording period was set to 24 h from 14:00 to 14:00 the next day. In total, 576 24-hour MSR files of 24 dairy goats and 24 dairy sheep were scheduled.

**2.5.2 Assessment of welfare issues, failure rate and data plausibility.** All animals were accustomed to wearing halters from previous experiments, and the halters were put on them for a minimum of 12 h prior to beginning recording for habituation. One set of halters (and their integrated data loggers) was used for the sheep and another set for the goats because it facilitated the individual adjustment of the halters to the head shape of the animals. Any behavioural problems (horns of other animals or barn furniture tangled in the halter, changed behaviour due to the halters) or lesions (hairless spots, wounds) due to the halters were noted daily after their removal.

It was noted daily, if a recording was not executed due to human error or animal welfare issues. Furthermore, all technical failures during the feeding experiment from all three MRs were recorded. These were identified by the number of MSR files that were not readable by Viewer2.

Thereafter, the Viewer2 output data files were analysed for implausibility values that would compromise data quality. It was assumed that data files of extreme values for feeding and/or ruminating duration would indicate incorrect classifications in these specific files. For the purpose of the experimental approach of this feeding experiment, data files with feeding and ruminating durations of less than 3.1 h/d were deemed implausible. These cut-off values were set from the minimal durations for feeding in goats reported by Lu [26] and for ruminating in small ruminants reported by Penning et al. [27]. Due to the lack of suitable literature, implausible high durations for feeding and ruminating were identified by outliers (+/-1.58 interquartile range/square root (n), [28]), which excluded feeding and ruminating durations exceeding 8.1 h/d and 10.7 h/d, respectively.

**2.5.3 Estimation of variance of data loggers.** Further, the order of magnitude of the variance attributed to the individual data loggers (n = 18) was estimated. Three output variables of Viewer2, the feeding duration in hours per day, rumination duration in hours per day and mastications per cud (MpC), were analysed each in a linear mixed effects model (lme4 package [24]) to estimate the variance of the random effect data logger. Other random effects for comparison included in the model were individual, species, the feed (HH, GH, MG) and the date. As the halters were separated into a set for sheep and a set for goats, the random effect of the data logger was nested within each species.

- Output variable ~ (1|feed) + (1|individual) + (1|species/logger) + (1|date)

## 3 Results

### 3.1 Comparison of the behaviour recording methods

In a descriptive analysis, all seconds of behaviour recording (n = 345,657) were compared between the recording methods (observation and Viewer2 classification). For all seconds that had been observed as ruminating, Viewer2 classified 74.2% of these correctly. For all seconds that had been observed as feeding, Viewer2 classified 91.7% of these correctly. The seconds at which no behaviour or other behaviour was observed and the software should "ignore" were correctly classified as "no or other behaviour" in 89.5% of seconds (Table 3).

### 3.2 Parameters of agreement

No difference was found between sheep and goats for the three parameters (S1 Table 3 in S1 Table): accuracy ($\chi^2(1)$ = 1.63, p = 0.20), specificity ($\chi^2(1)$ = 0.00, p = 0.99) and precision ($\chi^2(1)$ = 1.14, p = 0.29). Sensitivity was +0.06 higher in sheep than in goats ($\chi^2(1)$ = 6.06, p = 0.01). As this difference was rather small, the results of Viewer2 performance are presented on models with species as a random effect instead as a fixed effect.

For feeding behaviour, the estimated mean for accuracy, sensitivity, specificity and precision was 0.74 or higher. For ruminating behaviour, the estimated mean (± standard error) for sensitivity was above 0.72 (± 0.04), and above 0.97 (± 0.06) for accuracy, specificity and precision. Viewer2 was -0.03 less accurate ($\chi^2(1)$ = 8.71, p < 0.01) and -0.20 less sensitive ($\chi^2(1)$ = 46.35, p < 0.001) for ruminating than feeding (feeding: accuracy = 0.89 ± 0.04; sensitivity = 0.92 ± 0.04). It was +0.23 more specific ($\chi^2(1)$ = 47.08, p < 0.001) and +0.16 more precise ($\chi^2(1)$ = 34.44, p < 0.001) in its classification for ruminating than for feeding (feeding: specificity = 0.74 ± 0.08; precision = 0.81 ± 0.06; Fig 3).

**Table 3. Percentages of correctly and incorrectly classified seconds of Viewer2 compared with the observed behaviour of sheep and goats on pasture and in the pen at 10 Hz and 20 Hz (both species presented together as no relevant difference could be statistically detected in the analyses).**

| Seconds observed as . . . | | Pasture, 10 Hz | Pen, 10 Hz | Pen, 20 Hz | Sum |
|---|---|---|---|---|---|
| **. . . ruminating** | | | | | |
| **Number** | | 49,840 | 26,536 | 24,346 | 100,722 |
| **Percent classified** | correctly | 74.8 | 66.8 | 81.0 | 74.2 |
| | as feeding | 7.8 | 15.8 | 2.3 | 8.6 |
| **Percent** | not classified | 17.5 | 17.4 | 16.6 | 17.4 |
| **. . . feeding** | | | | | |
| **Number** | | 76,215 | 19,001 | 21,039 | 116,255 |
| **Percent classified** | correctly | 92.8 | 88.7 | 90.2 | 91.7 |
| | as ruminating | 0.5 | 2.6 | 0.3 | 0.8 |
| **Percent** | not classified | 6.7 | 8.7 | 9.5 | 7.5 |
| **. . . no behaviour** | | | | | |
| **Number** | | 13,591 | 46,753 | 56,454 | 116,798 |
| **Percent classified** | correctly | 81.5 | 96.3 | 93.3 | 93.1 |
| | as ruminating | 1.1 | 0.3 | 1.3 | 0.9 |
| | as feeding | 17.4 | 3.3 | 5.3 | 6.0 |
| **. . . other oral behaviour** | | | | | |
| **Number** | | 2,617 | 4,767 | 4,150 | 11,534 |
| **Percent classified** | correctly | 40.5 | 56.1 | 61.7 | 54.6 |
| | as ruminating | 1.9 | 1.9 | 5.0 | 3.0 |
| | as feeding | 57.6 | 42.0 | 33.3 | 42.4 |
| **. . . drinking** | | | | | |
| **Number** | | 158 | 150 | 220 | 528 |
| **Percent classified** | correctly | 10.1 | 32.7 | 86.4 | 48.3 |
| | as ruminating | 0.0 | 0.0 | 0.0 | 0.0 |
| | as feeding | 89.9 | 67.3 | 13.6 | 51.7 |

**3.2.1 Effect of condition: Pasture versus barn (10 Hz).** Accuracy was on a similar level for feeding and ruminating in the barn, but on pasture, it was -0.07 ($\pm$ 0.01) lower for ruminating than for feeding (S1 Table 3 in S1 Table, interaction behaviour x condition: $\chi^2 = 8.22$, p = 0.004). Sensitivity was -0.22 ($\pm$ 0.03) lower for ruminating than for feeding ($\chi^2 = 47.37$, p < 0.001) and -0.04 ($\pm$ 0.04) lower in the barn than on pasture ($\chi^2 = 4.65$, p = 0.03), with no interaction between the two factors ($\chi^2 = 0.85$, p = 0.356). Specificity for ruminating was similar for the two conditions (pasture: 0.95 $\pm$ 0.03; barn: 0.98 $\pm$ 0.04), but for feeding, it was +0.25 ($\pm$ 0.05) higher in the barn than on pasture (interaction behaviour x condition: $\chi^2 = 10.44$, p < 0.01). Precision for ruminating was similar between the conditions (pasture: 0.98 $\pm$ 0.02; barn: 0.97 $\pm$ 0.03), but for feeding, it was -0.23 ($\pm$ 0.03) lower in the barn than on pasture (interaction behaviour x condition: $\chi^2 = 20.76$, p < 0.001).

**3.2.2 Effect of recording frequency: 10 Hz versus 20 Hz.** Increasing the recording frequency from 10 Hz to 20 Hz improved the classification accuracy by +0.05 (S1 Table 3 in S1 Table; $\pm$ 0.02; $\chi^2 = 15.16$, p < 0.001), specificity by +0.07 ($\pm$ 0.02; $\chi^2 = 4.76$, p = 0.03) and sensitivity by +0.02 ($\pm$ 0.06; $\chi^2 = 3.16$, p = 0.08), with varying degrees of statistical certainty. For precision, no effect of recording frequency could be found ($\chi^2 = 1.20$, p = 0.27).

### 3.3 Application of JAM-R in a feeding experiment

**3.3.1 Values of feeding and ruminating behaviour.** On average ($\pm$ SD), sheep consumed 1.34 ($\pm$ 0.13) kg DM in 4.9 ($\pm$ 1.0) h/day, ruminated for 7.2 ($\pm$ 1.4) h/day and had 70.43 ($\pm$

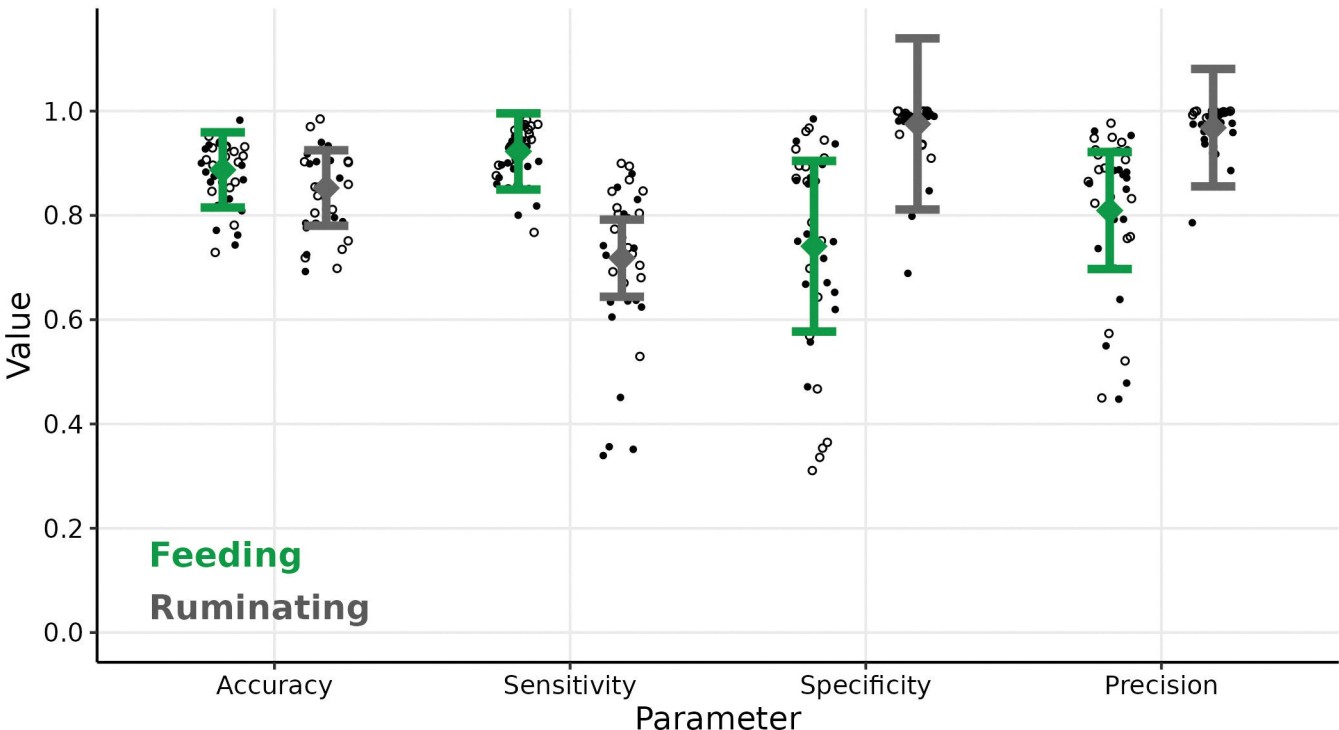

**Fig 3. Estimated mean (square with error bar of 95% confidence interval) parameters of agreement of Viewer2 for feeding and ruminating with observed values of goats (dots) and sheep (circles).**

10.7) MpC across the three MRs. The goats consumed on average 1.01 (± 0.14) kg DM in 4.9 (± 0.8) h/day, ruminated for 5.1 (± 1.1) h/day and had 63.9 (± 9.0) MpC. The distribution of feeding and ruminating (minutes per hour) over 24 h for the sheep and goats on the three MRs is shown in Fig 4.

**3.2.2 Assessment of welfare issues, failure rate and data plausibility.** Over the course of the entire experiment (October 2019 to April 2020), 8 of the initial 16 data loggers (50%) had to be replaced due to technical failure. These loggers were produced in 2009 or 2010 (used by Patt et al. [21]).

Only one goat managed to remove her halter once, but never again thereafter. It could not be documented whether the removal was undertaken by herself or her partner animal, but no injuries to either animal were found. Sheep had to learn how to enter through the wooden palisades with the data logger pocket in the beginning, but all managed within the second or third attempt and from the second day onwards, this struggle was rarely seen. Three goats and three of the four East-Frisian sheep developed (non-permanent) hairless spots where the noseband rested towards the end of the experiment of all three MRs.

From the 576 scheduled MSR files, in 14 cases (2.4%), the MSR file was missing due to human error (data loggers were unintentionally not programmed) or animal health (one animal could not wear the halter for the entire experiment due to an abscess on the jaw, which ruptured when trying on the halter the first time and then needed to heal), resulting in the analysis of 562 24-hour MSR files.

From the 562 MSR files executed, Viewer2 could not produce 52 (9.3%) output data files due to the technical failure of the data loggers. Seventeen of these were derived from two faulty loggers. If the recordings had been checked right away (processed by Viewer2 directly after the

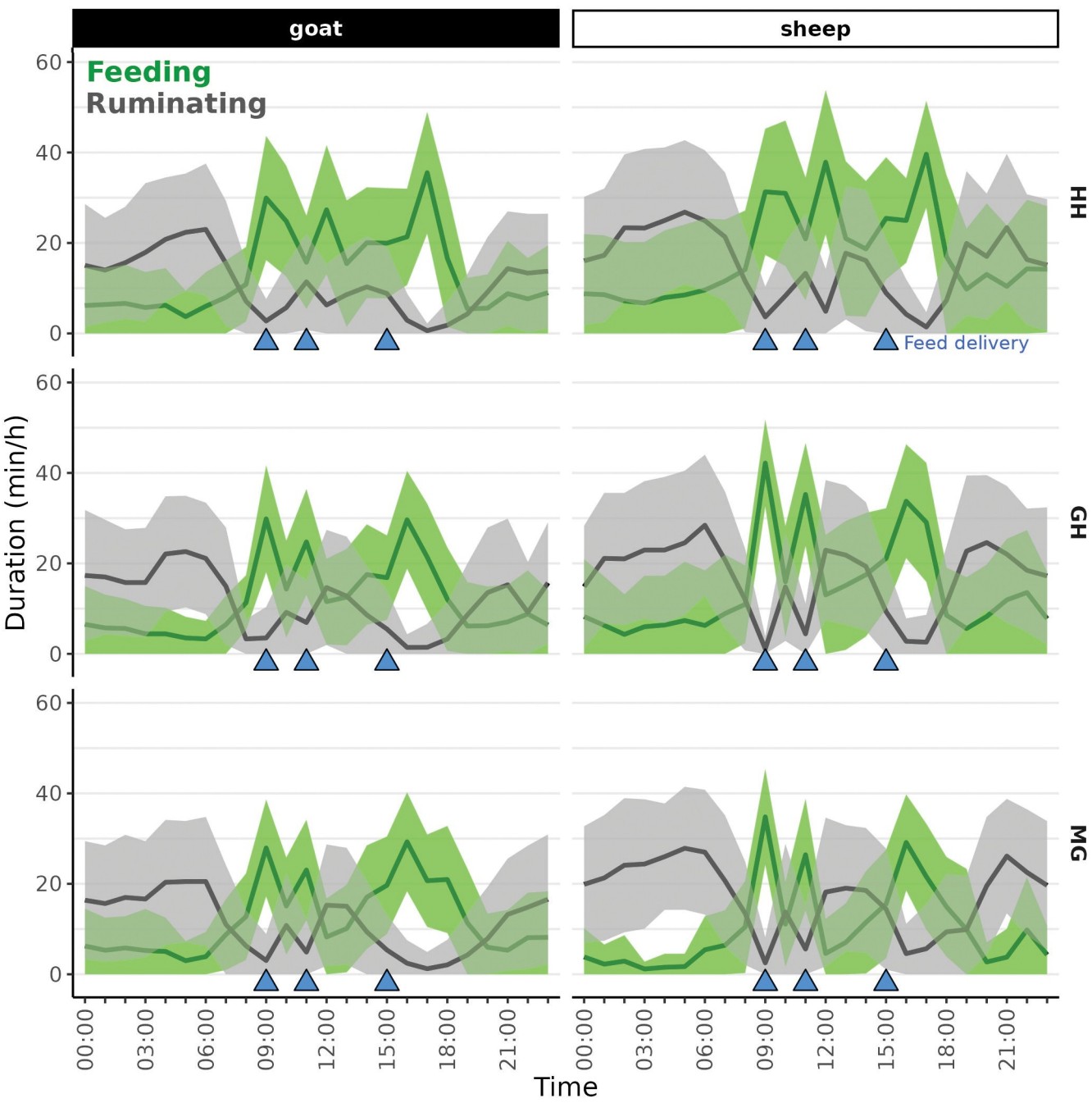

**Fig 4. Distribution of feeding and ruminating (mean ± standard deviation (shaded area)) of 24 sheep and 24 goats on three different mixed rations (HH, GH and MG, Table 1) over 24 h, with feed delivery (blue triangles) at 09:00, 11:00 and 15:00.**

recording was finished), the data loggers could have been replaced earlier, resulting in a possible missing recording rate of only 6.6%.

From the remaining 510 MSR files that were analysed by Viewer2, 109 (21.4%) output data files were identified to have implausible high (feeding: 23; ruminating: 0) or low (feeding: 25; ruminating: 81) durations of feeding and ruminating, resulting in a final dataset of 401 data files. This represents 71.4% of the executed 562 MSR files.

**Table 4. Variance of random effects of the linear mixed effects models of three Viewer2 output variables without fixed effects and listed random effects; Significance level of the random effect 'Logger' through model comparison.**

| Viewer2 output variable | | Rumination duration (h/d) | Feeding duration (h/d) | Mastications per cud (no.) |
|---|---|---|---|---|
| Random effect | n | Variance | Variance | Variance |
| Individual | 50 | 0.53 | 0.18 | 70.5 |
| Logger | 18 | 0.09 | 0.17 | 3.5 |
| Species | 2 | 2.69 | 0.00 | 21.2 |
| Feed | 3 | 0.00 | 0.20 | 1.4 |
| Date | 36 | 0.03 | 0.01 | 1.0 |
| Residual | | 0.93 | 0.51 | 40.0 |

**3.3.3 Variance of data loggers on output variables of Viewer2.** The variance of the random effects from the models to estimate duration of rumination, duration of feeding and MpC are shown in Table 4. The variance of the data loggers was in the range of the variance of other random effects.

## 4 Discussion

By recording feeding and ruminating behaviour on pasture and in the barn, JAM-R was validated for sheep and goats. The classification performance of the Viewer2 software was assessed by behaviour observation. By using the system in a feeding experiment, issues during the application, such as technical failure rates and welfare issues, were assessed.

With a classification accuracy of more than 0.85 across species, recording conditions and frequency, Viewer2, which was developed for dairy cows [12], can be reliably used to monitor the feeding and ruminating behaviour of sheep and goats. For monitoring daily feeding and ruminating, an overall accuracy of more than 0.85 at 10 Hz already provides a good estimate of these behaviours for dairy sheep and goats under the conditions investigated. However, an increase in recording frequency from 10 Hz to 20 Hz slightly improved the classification performance of Viewer2 in this study. For a more detailed analysis of ruminating and feeding behaviour, and depending on the research question, a recording frequency of 20 Hz (or even higher) could therefore be advisable. A higher recording frequency could also be more accurate in detecting single mastications, but this was not evaluated in this study. Notably, a higher recording frequency also produces a larger amount of data, which entails increased processing time for saving files and for calculations with Viewer2. The effect of the recording frequency on the parameters of agreement needs to be tested under additional conditions, such as pasture.

Some lower classification values of the presented system might be explained by the methodological shortcomings of the human observer. During observations, the definition of ruminating included chewing and regurgitation of cuds, as pauses in between cannot be reliably observed. An observation therefore started with the first chewing movements and ended after no further cud was regurgitated, including the seconds of swallowing and regurgitation between cuds. In contrast, Viewer2 classified only the seconds of actual chewing during the rumination and excluded the time of swallowing and regurgitation between cuds. This difference in time is reflected in the lower sensitivity.

Additionally, the high variation and comparably low values for classification specificity on pasture compared with the recordings in the barn could be caused by a methodological difference between live and video observation. The animals in the experimental pens were continuously video recorded for three hours, during which all behaviours were recorded, including many seconds of other activities apart from feeding and ruminating. On pasture, the observer

actively switched to individuals demonstrating feeding or ruminating. Therefore, observations of no behaviour or behaviour other than feeding and ruminating were underrepresented. As the calculation for specificity is based on these *true negative* seconds, in which no or other behaviour is correctly classified as such, there might not have been sufficient data to calculate reliable values for this parameter on pasture. To cover the full potential of the JAM-R, further aspects need to be investigated. For example, the classification performance in recording periods, where feeding in the barn and on pasture must be distinguished, or the applicability in natural environments, where the predominant feeding activity is browsing rather than grazing.

Based on the same pneumatic system and similar classification software, the commercially available system RumiWatch has been validated in recent years by several research groups and has shown performance values similar to those in our study [29–31]. Other systems for identifying feeding and ruminating behaviour use the position of the head and/or the mouth of the animals, recorded by 3-axial [32] or 1-axial accelerometers [33]. These systems show performances similar to Viewer2 in classifying grazing behaviour, but are much less reliable in identifying rumination (precision: 0.5–0.8, sensitivity: 0.1–0.7; [32]) or are not applicable to this behaviour [33]. Additionally, these systems can only be used for grazing animals, as the downward head position while moving forward is a characteristic of feeding on pasture but not in the barn.

Scarce literature reports all-day ruminating and feeding times, especially for small ruminants. For pasture systems, live or video behaviour observations during the night are generally limited (as reported in e.g. [17,34]) and miss the majority of ruminating behaviour, which mostly takes place during the night [35,36]. This demonstrates an advantage of the automatic recording system, as it can be used independently of housing and feed type. Feeding and ruminating durations indoors can vary significantly, although the values obtained in our study are within the range reported in the literature. Feeding duration can be as low as 3.1 h/day for goats [26] and 1.7 h/day for sheep [37] but can also rise to 13.3 h/day [Bell 1957 as cited in: 35]) and 8.0 h/day [38] for goats and sheep, respectively. Ruminating durations, when sheep and goats are kept indoors, have been reported to be 8.7 h/day and 7.2 h/day, respectively, on a straw and hay diet [3]. The number of mastications or chews for cows, monitored with the cow version of the JAM-R and fed with hay, has been reported with a mean of 52 ± 14 MpC [13,39]. The number of mastications of small ruminants has rarely been reported. According to Jalali et al. [3], the mean MpC was 69 MpC and 74 MpC for sheep and goats, respectively, which is similar to the values found in the present study.

Technical failure rate, information on missing data or issues in relation to fitting the devices to the animals are almost never reported in validations or presentations of similar technologies, as criticised by Rombach et al. [30]. The technical failure rate of 21% for the RumiWatch system in their study was similar to the present study. Even though only 71.4% of the executed data set could be used in the final data of the feeding experiment, the JAM-R seems to be a reliable behaviour recording tool. Some of the technical failures could have been minimised with the experience gained during the application of the system. Daily checking of the functioning of the data logger files with Viewer2 before re-adjustment of the halters could have reduced the failure rate considerably.

Furthermore, the animals' habituation to wearing a head collar and the fit of the head collar seem to be crucial for recording quality [9] and welfare. As each data logger was associated with one halter, at least part of the data logger variance reported for the durations of feeding, ruminating and the MpC could be attributed to the fit of the halter. When the head collar fits too loosely, the device will not sufficiently detect differences in pressure. When the head collar is too tight, pain and friction caused by the halter when the jaw moves could reduce feeding

and ruminating. Head collars that are size adjustable at multiple points, so that they can be individually fitted to the animal's head shape, are recommended to support recording quality and welfare.

When comparing suitability for research, the JAM-R is more complicated but more flexible than commercial devices. For instance, Viewer2 needs reference sequences of the behaviours for each file (i.e. per animal and recording period), whereas the RumiWatch system for cows can transfer its learning sequence [30], but this has only been shown for cattle. An advantage of the JAM-R is that user settings, similar to recording frequency, can be set manually. Furthermore, the data loggers can be programmed to additionally record 3-axial acceleration, which can be integrated in the identification of feeding on pasture [32]. This flexibility and the potential for further development make the JAM-R system especially interesting for research. There is even the potential to adapt this methodology to other ruminating species in captivity, where it might be important that the recording frequency can be adapted depending on the behaviour under investigation.

In conclusion, according to the values of accuracy, sensitivity, specificity and precision, the presented JAM-R system with the calculations of Viewer2 provides a reliable and applicable technology for automatic recording of feeding and ruminating behaviour of small ruminants on pasture and in the barn. A recording frequency of 10 Hz has proven to be sufficient but can be adapted according to the research question.

## Supporting information

**S1 Table. Table 1: Table of components and the chemical composition per kilogram dry matter (DM) of the three mixed rations: First- and second-cut hay (HH), grass silage and hay (GH), and maize silage and grass silage (MG).** Table 2: Descriptions of true and false positive and negative seconds of feeding and ruminating. Table 3: Model estimates of the agreement parameters and model comparison results for significance of fixed effects.
(DOCX)

**S1 File. Supplementary 2 method description analysis of feed samples.**
(DOCX)

**S1 Data. Supplementary 3 Data 1.**
(XLSX)

**S2 Data. Supplementary 4 Data 2.** Raw data of method comparison per individual and condition.
(XLSX)

## Acknowledgments

We thank the technicians and animal care staff at Agroscope Tänikon for their help in conducting the experiment. We are also grateful to Sébastien Dubois and his team from the Analytic Department of Agroscope Posieux for the chemical analysis of the feed. Additionally, we express our appreciation to Joël Berard (head of the Animal Production Systems and Animal Health Department, Agroscope), Katharina Zipp and Antonia Patt for introducing us to the use of MSR data loggers and JAM-R.

## Author Contributions

**Conceptualization:** Roxanne Berthel.

**Data curation:** Roxanne Berthel, Alisha Deichelboher.

**Formal analysis:** Roxanne Berthel, Alisha Deichelboher.

**Funding acquisition:** Nina Keil.

**Methodology:** Roxanne Berthel.

**Project administration:** Frigga Dohme-Meier, Nina Keil.

**Resources:** Frigga Dohme-Meier, Wendelin Egli, Nina Keil.

**Software:** Wendelin Egli.

**Supervision:** Nina Keil.

**Visualization:** Roxanne Berthel.

**Writing – original draft:** Roxanne Berthel.

**Writing – review & editing:** Roxanne Berthel, Alisha Deichelboher, Frigga Dohme-Meier, Wendelin Egli, Nina Keil.

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
