## [Decision Letter · Decision Letter 0]

15 Dec 2022

PONE-D-22-30895Automatic monitoring of feeding and ruminating behaviour of sheep and goatsPLOS ONE

Dear Dr. Berthel,

Thank you for submitting your manuscript to PLOS ONE. After careful consideration, we feel that it has merit but does not fully meet PLOS ONE’s publication criteria as it currently stands. Therefore, we invite you to submit a revised version of the manuscript that addresses the points raised during the review process. Dear Authors

I believe that the manuscript is well prepared, however, I have concern regarding for the JAM-R recording as also suggested by one of the reviewer in feeding pens, a one day recording for 10 and 20 Hz recoding was analyzed. Is there any pervious analysis that may suggest the one day recording can generate sufficient data for analyzing the feeding and ruminating behavior in pen situation? While analyzing the application data, at what recording frequency was the data summarized (min, sec, hr?) and was that treated as a repeated measure in the model for analyses? Therefore, i suggest major revision and request to address the comments raised by reviewers.

We look forward to receiving your revised manuscript.

Kind regards,

Aziz ur Rahman Muhammad

Academic Editor

PLOS ONE

Journal Requirements:

4. We note that Figure 1 in your submission contain copyrighted images. All PLOS content is published under the Creative Commons Attribution License (CC BY 4.0), which means that the manuscript, images, and Supporting Information files will be freely available online, and any third party is permitted to access, download, copy, distribute, and use these materials in any way, even commercially, with proper attribution. For more information, see our copyright guidelines: http://journals.plos.org/plosone/s/licenses-and-copyright.

5. We note that Figure S1 in your submission contain map/satellite image which may be copyrighted. All PLOS content is published under the Creative Commons Attribution License (CC BY 4.0), which means that the manuscript, images, and Supporting Information files will be freely available online, and any third party is permitted to access, download, copy, distribute, and use these materials in any way, even commercially, with proper attribution. For these reasons, we cannot publish previously copyrighted maps or satellite images created using proprietary data, such as Google software (Google Maps, Street View, and Earth). For more information, see our copyright guidelines: http://journals.plos.org/plosone/s/licenses-and-copyright.

a. You may seek permission from the original copyright holder of Figure S1 to publish the content specifically under the CC BY 4.0 license.  

Additional Editor Comments:

Dear Authors

I believe that the manuscript is well prepared, however, I have concern regarding for the JAM-R recording as also suggested by one of the reviewer in feeding pens, a one day recording for 10 and 20 Hz recoding was analyzed. Is there any pervious analysis that may suggest the one day recording can generate sufficient data for analyzing the feeding and ruminating behavior in pen situation? While analyzing the application data, at what recording frequency was the data summarized (min, sec, hr?) and was that treated as a repeated measure in the model for analyses?

Therefore, i suggest authors major revision and address the comments raised by reviewers

Reviewers' comments:

Reviewer's Responses to Questions

**Comments to the Author**

1. Is the manuscript technically sound, and do the data support the conclusions?

Reviewer #1: Partly

Reviewer #2: Yes

Reviewer #3: Yes

2. Has the statistical analysis been performed appropriately and rigorously? 

Reviewer #1: Yes

Reviewer #2: Yes

Reviewer #3: Yes

3. Have the authors made all data underlying the findings in their manuscript fully available?

Reviewer #1: Yes

Reviewer #2: No

Reviewer #3: Yes

4. Is the manuscript presented in an intelligible fashion and written in standard English?

Reviewer #1: No

Reviewer #2: Yes

Reviewer #3: No

5. Review Comments to the Author

Reviewer #1: General comments:

The authors set out to validate the JAM-R technology for monitoring feeding and rumination behavior sheep and goats, based on previously successful findings on dairy cows. Any precision technology is an interesting topic which will most certainly be implemented in most domesticated species in the future to some extent. Additional knowledge regarding how to best quantify feeding and ruminating behaviors in ruminants is important and will help assess both general population health and welfare status if validated and implemented properly. I was curious about the manuscript findings from the get-go. Accuracy, sensitivity, specificity, and precision of the JAM-R technology were reported by authors as moderately to highly correlated between the JAM-R technology and human observations for both goats and sheep, indicating that they may be useful tool for both species under commercial conditions.

Although this paper certainly has merit, some restructuring and condensation of text as well as proof-reading are needed to improve readability and clarity. Additional clarifications are needed across the material and method and statistical analyses. In addition, one of the sub-sections (MR trial) of this manuscript is currently not methodologically aligned to the overarching aim of validating the JAM-R. Additional data (if available) could pull this together to fit nicely with the rest of the manuscript. That said, I think this is an important field of work with great utility.

Some specific comments as line by line below:

Title: Since the manuscript is mainly focusing on validating the JAM-R technology, I think this should be clarified in the manuscript title. Perhaps along the lines of: “Validation of feeding and rumination behaviours using an automatic jaw-movement recording system in sheep and goats.”

Abstract:

Overall, the abstract reads well. In light of my comments below, lines 36-38 may need to be omitted as they currently are not properly linked to the overarching aim of the paper. I think that the paper is mainly focused on validating the technology more so than describing sheep and goat feeding behavior which as been done previously. The novelty lies in the validation aspect.

Introduction:

Ln 79-80: Change to past tense. E.g. ‘The JAM-R technology has previously been validated for dairy cows’.

Ln 102-103: Consider rewording ‘checked’ to ‘investigated’. Also, I find this last aim of the paper somewhat strange. A better aim would be to investigate if the JAM-R technology is consistently accurate and precise between commonly used goat and sheep MR’s, and then compared the values to the existing literature regardless. As it currently sits, it’s purely descriptive and a stand-alone-aim not linked to validating the JAM-R technology. If you have additional data for the feed trial, I would highly encourage to analyze this and add it to the paper.

Material and methods:

Overall, the material and method reads a bit jumbled and could need some clarification and separation between methods, data editing and statistics.

Ln 122-125: I would consider moving the breed information up after the first sentence of this section (ln 116).

Ln 147: Redundant line, information given previously.

Ln 142-158: I am a bit uncertain of the numbers of replicates used. Five or eight? But 1 replicate for animals on pasture? Please clarify.

Ln 168: This is an unfortunate limitation of the study. It would have been great to have several observers and their inter-reliability scores presented to reduce the risk of bias regardless of the experience of the observer, especially as the ethograms differed slightly between species. Was any intra-reliability score conducted?

Ln 167-241: This section of the manuscript is a bit hard to follow. Consider making a main header named “Statistical analysis” for only the statistical approach and make an appropriate sub-headers for the accuracy, sensitivity, specificity and precision formulas. Perhaps the sub-header could be simply: ‘Validation formulae”?

The other information should be restructured and placed in material and methods under a sub-header such as ‘‘Data classification and management” or similar for ease of flow and readability.

Ln 285-299: This section could use some clarification, please spell out the parameters and factors used (or the LME formula) in each significant step to help the reader follow the path of analysis.

Ln 300: Consider renaming this section as the second project/trial and move this section up before the statistical analysis section.

Ln 326: If these MR’s are commonly used, perhaps the full chemical analyses section may be redundant in the scope of the paper and could be omitted. As mentioned earlier, if the JAM-R was tested on different MR’s rather than only providing descriptive data on the different MR’s per species, that would be better aligned with the scope of the paper.

Ln 342 and elsewhere: N= 345,657. Change to a comma for all numbers except for the decimals ‘.’.

Results:

Rename the headers to reflect the structure in the manuscript after suggested changes above.

Ln 356: What statistical analysis was done here? A chi-square? Please clarify in the statistical analysis section. Currently only LME and ANOVA are listed.

Ln 363: are the (+/-) in SE or SD? Please clarify.

Ln 389-410: This section is presenting data not specifically related to the main aim or objective of the paper. The presentation of the estimated feeding behaviours for each species is somewhat redundant when not compared to manual observation. As it stands, the results presented are bound to be more linked to housing and feed rather than the technology.

Discussion:

Overall, I think the discussion reads nicely with the study limitations elaborated on.

Ln 412: Here and throughout, I think it would better to refer to the validation process as to validating the whole system instead of the Viewer 2 software.

Ln 512: Change reference header to English.

Figures and tables:

Figure 2: My recommendation would be to edit this figure with new headers in English to avoid a lengthy figure legend with translations.

Figure 4: Although these findings have merit and are interesting in it’s own right, this figure does not contributed to answering any of the aims set out in the beginning the manuscript unless compared to manual observation. As it stands, the results presented are bound to be more linked to housing and feed rather than the technology.

Table 5: I don’t know if this particular table is of value in the light of the scope of the manuscript unless the difference between the technology estimation and true consumption is presented. The difference in consumption between species seem like a stand-alone result outside of the current aims. As it stands, the results presented are bound to be more linked to housing and feed rather than the technology.

Reviewer #2: This study is interesting and valuable for investigation of animal behavior and other fields. However, it is not suitable for acceptation in the present form because of the important issues followed:

(1) The authors did not clearly describe the procedure for this investigation sequently in Method, especiallly for the acheivement and processing of the raw data. If possible, why not present a overview of the data processing? In addition, I am also not clear how the Viewer2 work in the present study in this version of manuscript. Is it just a software for data statistic analysis? I think the authors should make some explanation about the Viewer2 in this respect.

(2)The aim of this study seems to validate the performance of the Viewer2 software in classifying the feeding and ruminating behavior in sheep and goat, using the observation by video as the standdard for caculation of the efficiency in the system of JAM-R and Viewer2. However, I could not find comparison betweenthe two methods in context. In addition, the data from video observation should also be displayed in supplementary files.

Further more, there were some errors in grammar, i.e. line 257-258, as well as formats of typings in the manuscript.

Reviewer #3: This review is for the manuscript titled “Automatic monitoring of feeding and ruminating behaviour of sheep and goats”.

Major comments

Naturally, goats are browsers and sheep are grazers where their natural grazing behavior can be assessed. Can the authors provide the reason why they used goats on pasture grazing to evaluate feeding behavior study which may not reflect the natural behavior of the animal, especially under grazing condition?

For the JAM-R recording in feeding pens, a one day recording for 10 and 20 Hz recoding was analyzed (L198-99). Is there any pervious analysis that may suggest the one day recording can generate sufficient data for analyzing the feeding and ruminating behavior in pen situation?

While analyzing the application data, at what recording frequency was the data summarized (min, sec, hr? Line 393) and was that treated as a repeated measure in the model for analyses?

Minor comments

Abstract

Line 39-41: review as “… system with the Viewer2 software provided a reliable technology for automatic recording of feeding and ruminating behavior of sheep and goats on pasture and in the barn”.

Introduction

The introduction section is a bit long and requires revision to make it concise and also increase its flow.

Line 45: revise as “… monitoring feeding and ruminating behavior of ruminants is …”

Lines 61-64: please revise this sentence is not clear

L65; … feeding and rumination ….

L69-73: this seems to be a single sentence and can be revised as “…(15) including; 1) …; 2)…..; and 3)…”

L74-76: revise as “… feeding and ruminating behaviours as the duration and frequency of mastication differs.”

L79: revise as “… (18). The system was tested and validated for dairy cows (19).”

M & M

L108: revise “To validate the softrware Viewer2, feeding and ….”

L116: delete respectively

L121-22: revise this sentences. Is this means 2 goats (from pasture) for observation, 5 goats (including the two) for video, and 5 sheep for video monitoring used? Also, clarify if the animals used for monitoring behavior on pasture and pen settings were the same.

L114-131: include the average body weight (±SD) of the experimental sheep and goats

L140: revise “… pend had three drinking water sources and one mineral supply site.”

L156: Is there a reason why the sheep were moved to the same plots that is already used by the goats earlier?

L164-65: just out of curiosity, what is the value of having a round table in the pens?

L165: revise as “ Experimental animals received a MR containing 55% …… ad libitum.”

L167: Naturally, goats are browsers and sheep are grazers where their natural grazing behavior can be assessed. Can the authors provide the reason why they used goats on pasture grazing to evaluate feeding behavior study which may not reflect the actual natural behavior of the animal?

L172-73: revise as “… on the weather condition, ….”.

During higher temperature, animals tend to be less active and gather under a shade/ hut and this is one of the occasions that rumination takes place. So, during grazing in the field, is the observation in the morning specifically focus on feeding and the one in the afternoon focused on rumination?

L177: which previous experiment? Include the source. This may also answer the question if the two days of recording can generate sufficient data for behavioral study. As indicated in L198, basically it is a one day data as different recording frequency (10 vs 20 Hz) was used on separate days.

L205: different font

L218: revise as “ …file (one per animal per day) ..”

L226: Based on L144 L160, and L301, all the grazing and pen feeding experiments were conducted in 2020 however, in the excel spreadsheet provided here (under application tub in the file) the date stamp indicated data collected in 2019.

L300: Is this a separate experiment from the one stated in L159-66 for pen feeding that used just five animals, March to April 2020?

L306-7: explain the reason for providing different MRs? Also, revise – a 14 days adaptation period was mentioned on L314 – so perhaps a specific MR was provided for a total of 24 days (14 adaptation and 10 experimental).

L324-5: what was the recording frequency considered for JAM-R (min, sec, hr? L393)and was that tread as a repeated measure in the model for analyses?

L326: please include the quality analysis results for the grazed pasture as well

Results

L341: The system seems a poor predictor for drinking (Table 4) especially on pasture. Include descriptions on drinking behavior as well.

L347: revise as “… were not correctly classified …”. As the system depends on change in pressure (with no visual aid to specifically identify and label activities), the observed

L373 & 384: include the details of the results reported under these sections in a table form, perhaps expanded on Table 4 or added as supplementary table.

L390: revise as “On average (±SD), sheep fed for 5.1….. MpC across the three MRs. Conversely, goats fed ….”

L403-4: On L303-4, stated that data was collected from 26 sheep and goats but only results from 24 is reported here.

L512: References

Tables

Table 5: Include the unit for dry matter intake

Supplementary 1 (word file)

The reported size of area and the numbers placed on the demarcated landscape doesn`t match. For example, for observation goats site the number on the map is 100 m by 119.55 m but the reported area 503.58 m2. The same for the sheep sites as well.

Supplementary (excel file)

As stated above, the data collection year needs to be checked (2019 vs 2020). The headings for individual columns within each tabs needs to be clearly presented/stated. For example, in the ‘application’ tab ‘starttime’ in column A is indicated as 140000 and in column J and K separate date and time reported.

6. PLOS authors have the option to publish the peer review history of their article (what does this mean?). If published, this will include your full peer review and any attached files.

Reviewer #1: No

Reviewer #2: No

Reviewer #3: No

---

## [Author Response · Author response to Decision Letter 0]

6 Mar 2023

Answer to the Editor

Thank you for your feedback and comments. We have revised the manuscript, taking careful account of all the reviewers comments.

According to your comments, we added the copyright information on the figures:

We added the copyright indication on figure 1 and 2 and we added the source to the picture Supplementary Figure 1.

With this information, we have the permission to use Google maps screenshots in journals, as you can read here: https://about.google/brand-resource-center/products-and-services/geo-guidelines/#google-maps

Answer to Reviewer #1

General comments

Answer: 

Thank you very much for your valuable feedback and recommending our work for publication. We have revised the manuscript, taking careful account of all your comments.

Some specific comments as line by line below:

Title: Since the manuscript is mainly focusing on validating the JAM-R technology, I think this should be clarified in the manuscript title. Perhaps along the lines of: “Validation of feeding and rumination behaviours using an automatic jaw-movement recording system in sheep and goats.”

(1) We changed the titel to “Validation of automatic monitoring of feeding behaviours in sheep and goats”.

(2) Abstract:

Overall, the abstract reads well. In light of my comments below, lines 36-38 may need to be omitted as they currently are not properly linked to the overarching aim of the paper. I think that the paper is mainly focused on validating the technology more so than describing sheep and goat feeding behavior which as been done previously. The novelty lies in the validation aspect.

Answer: 

We changed the focus of the feeding experiment and instead of reporting differences between sheep and goats we report number of technical failures and information on missing data. The main result of this assessment was instead included in the Abstract.

Lines 36-37: “Applying the JAM-R in a feeding experiment, 71 % of the recordings executed were defined as technically error-free and produced plausible values of feeding behaviours.”

Introduction:

(3) R1: Ln 79-80: Change to past tense. E.g. ‘The JAM-R technology has previously been validated for dairy cows’.

Answer: 

It now reads (lines 69-70): “The JAM-R technology has previously been tested and validated for dairy cows.”

(4) R1: Ln 102-103: Consider rewording ‘checked’ to ‘investigated’. Also, I find this last aim of the paper somewhat strange. A better aim would be to investigate if the JAM-R technology is consistently accurate and precise between commonly used goat and sheep MR’s, and then compared the values to the existing literature regardless. As it currently sits, it’s purely descriptive and a stand-alone-aim not linked to validating the JAM-R technology. If you have additional data for the feed trial, I would highly encourage to analyze this and add it to the paper.

Answer: 

The word “checked” was changed to “investigated” in line 90. 

We now emphasize on issues of applicability and issues with fitting the halter (see Revisions of 2.5. M&M and 3.3 Results). 

It now reads (lines 93-96): “Additionally, the system was applied to sheep and goats in a feeding experiment on three different mixed rations (MRs) to assess the plausibility of obtained values and report frequencies of technical failure and possible problems for the welfare of the animals due to the wearing of the halter.”

Material and methods:

(5) Overall, the material and method reads a bit jumbled and could need some clarification and separation between methods, data editing and statistics.

Answer: 

The Material and Methods and the results sections were reorganized and information added. Here you see the new table of content of the Material and Methods and the Results section including line numbers.

2 Material and Methods 7

2.1 Animals and housing 8

2.1.1 Non-experimental condition 8

2.1.2 Pasture 9

2.1.3 Experimental pens 10

2.1.4 Feeding experiment for application of JAM-R 11

2.2 Behaviour recording by observation 11

2.3 Automatic behaviour recording 13

2.3.1 MSR-jaw-movement recording system (JAM-R) 13

2.3.2 Viewer2 behaviour classification software 15

2.4 Comparison of the behaviour recording methods 16

2.4.1 Data management 16

2.4.2 Parameters of agreement 17

2.4.3 Effects of species, condition and recording frequency on agreement parameters 18

2.5 Application of JAM-R in a feeding experiment 20

2.5.1 Description of the feeding experiment 20

2.5.2 Assessment of welfare issues and data quality 21

2.5.3 Estimation of variance of data loggers 22

3 Results 22

3.1 Comparison of the behaviour recording methods 22

3.2 Parameters of agreement 24

3.2.1 Effect of condition: Pasture versus barn (10 Hz) 25

3.2.2 Effect of recording frequency: 10 Hz versus 20 Hz 25

3.3 Application of JAM-R in a feeding experiment 26

3.3.1 Values of feeding and ruminating behaviour 26

3.3.2 Assessment of welfare issues and data quality 26

3.3.3 Variance of data loggers on output variables of the Viewer2 27

(6) R1: Ln 122-125: I would consider moving the breed information up after the first sentence of this section (ln 116).

Answer: 

Breed information was moved in the paragraph (Lines 114-115).

Abbreviations for breeds names were introduced to later report the number of animals per breed: “The sheep were of the dairy breeds Lacaune (LC) and East-Friesian (EF). The goats were of the dairy breeds Saanen (SA), Chamois Coloured (CC) and hybrids of these two breeds (HY).”

(7) R1: Ln 147: Redundant line, information given previously.

Answer: 

The line was deleted.

(8) R1: Ln 142-158: I am a bit uncertain of the numbers of replicates used. Five or eight? But 1 replicate for animals on pasture? Please clarify.

Answer: 

The paragraph was rephrased for better understanding.

It now reads (lines 145-149): “For the observations of the goats on five consecutive days, the plot was extended by 10 m x 30 m (total plot = 10 m x 50 m). When the observation period of the goats was finished, the sheep were then moved to this plot, and it was extended by an additional 10 m x 10 m (total plot = 10 m x 60 m; Supplementary Figure 1). On this plot, the behaviour observation of sheep was conducted for six consecutive days.”

(9) R1: Ln 168: This is an unfortunate limitation of the study. It would have been great to have several observers and their inter-reliability scores presented to reduce the risk of bias regardless of the experience of the observer, especially as the ethograms differed slightly between species. Was any intra-reliability score conducted?

Answer: 

The ethograms for the two species were exactly the same (Table 1, line188). May you please state where you found any contrary information so we can correct this?

We included a paragraph on inter-observer reliability for the video analysis (lines 180-183): “Video behaviour observation was assessed for agreement with another trained observer (R.B.) for 21600 seconds of two videos. R.B. reached an agreement of 97.6 - 98.2% for seconds observed as feeding by A.D. and 98.5 - 99.9% for seconds of rumination of A.D..”

(10) R1: Ln 167-241: This section of the manuscript is a bit hard to follow. Consider making a main header named “Statistical analysis” for only the statistical approach and make an appropriate sub-headers for the accuracy, sensitivity, specificity and precision formulas. Perhaps the sub-header could be simply: ‘Validation formulae”?

The other information should be restructured and placed in material and methods under a sub-header such as ‘‘Data classification and management” or similar for ease of flow and readability.

Answer: 

M&M section was restructured and headers renamed. We hope it is now more easy to follow. Please see point (5) for the table of contents.

(11) Ln 285-299: This section could use some clarification, please spell out the parameters and factors used (or the LME formula) in each significant step to help the reader follow the path of analysis.

Answer: 

We added model parameters and formulas in the section (line 305 – 335)

(12) Ln 300: Consider renaming this section as the second project/trial and move this section up before the statistical analysis section.

Answer: 

The section was renamed as “2.5 Application of JAM-R in a feeding experiment” and subheadings were introduced with explaining the (new) purpose of this section.

(13) Ln 326: If these MR’s are commonly used, perhaps the full chemical analyses section may be redundant in the scope of the paper and could be omitted. As mentioned earlier, if the JAM-R was tested on different MR’s rather than only providing descriptive data on the different MR’s per species, that would be better aligned with the scope of the paper.

Answer: 

The feed analyses and values had been moved to the supplementary (Supp. Table 1 and Supp. Methods Description).

(14) Ln 342 and elsewhere: N= 345,657. Change to a comma for all numbers except for the decimals ‘.’.

Answer: 

done.

Results:

Rename the headers to reflect the structure in the manuscript after suggested changes above.

Answer: 

The headers were renamed. See point (5) for the table of contents.

(15) Ln 356: What statistical analysis was done here? A chi-square? Please clarify in the statistical analysis section. Currently only LME and ANOVA are listed.

Answer: 

The R function anova () does not perform an ANOVA. 

It is now explained in the methods section (lines 331-334): “The models were stepwise reduced by the interaction and then by each fixed effect and compared with the full model. This was done using the anova function (25), which performs a χ2 test based on the restricted maximum likelihood method to determine whether the fixed effect omitted in the reduced model is statistically significant for the model fit.”

(16) Ln 363: are the (+/-) in SE or SD? Please clarify.

Answer: 

It is the standard error. The missing information was added in line 402.

(17) Ln 389-410: This section is presenting data not specifically related to the main aim or objective of the paper. The presentation of the estimated feeding behaviours for each species is somewhat redundant when not compared to manual observation. As it stands, the results presented are bound to be more linked to housing and feed rather than the technology.

Answer: 

The focus and reported data of this section was changed in relation to several issues of applicability. It now also reports reasons and frequencies of missing data, any technical errors during the application and potential problems arising with the fit of the halters. The focus of the statistical model is now about magnitude of the random effects, especially the variation explained among loggers. The section header was renamed (line 430) “3.3 Application of JAM-R in a feeding experiment”. The headers of the subsections are now (line 431) “3.3.1 Values of feeding and ruminating behvaiour”, (line 442) “3.3.2 Assessment of welfare issues and data quality” and (line 462) “3.3.3 Variance of data loggers on output variables of the Viewer2”.

Discussion:

Overall, I think the discussion reads nicely with the study limitations elaborated on.

(18) Ln 412: Here and throughout, I think it would better to refer to the validation process as to validating the whole system instead of the Viewer 2 software.

Answer: 

The first paragraph of the discussion was adjusted to cover both parts of the JAM-R validation and assessment. 

It now reads (lines 472-475): “By recording feeding and ruminating behaviour on pasture and in the barn, JAM-R was validated for sheep and goats. The classification performance of the Viewer2 software was assessed by behaviour observation. By using the system in a feeding experiment, issues during the application, such as technical failure rates and welfare issues, were assessed.”

Addition: Paragraphs 538-555 were adjusted to discuss the results of the new focus of the assessment of the application. The paragraph discussing the comparing values of feeding and rumination durations was shortened.

(19) Ln 512: Change reference header to English.

Answer: 

done.

Figures and tables:

(20) Figure 2: My recommendation would be to edit this figure with new headers in English to avoid a lengthy figure legend with translations.

Answer: 

Figure 2 is now presented in english. 

(21) Figure 4: Although these findings have merit and are interesting in it’s own right, this figure does not contributed to answering any of the aims set out in the beginning the manuscript unless compared to manual observation. As it stands, the results presented are bound to be more linked to housing and feed rather than the technology.

Answer:

The focus of the application experiment was changed in relation to further issues of applicability. However from our point of view, the issue of plausibility of the output data provided by the Viewer2 (summary of feeding behavior per day and distribution over day) is part of that question. We therefore would argue for keeping Figure 4. 

(22) Table 5: I don’t know if this particular table is of value in the light of the scope of the manuscript unless the difference between the technology estimation and true consumption is presented. The difference in consumption between species seem like a stand-alone result outside of the current aims. As it stands, the results presented are bound to be more linked to housing and feed rather than the technology.

Answer: 

Some of its information was transferred into the text (lines 432-434) to allow for evaluating the plausibility and comparison of its values with the literature in the discussion. The table is deleted. A new table 5 now presents the variance of random effects, including the data logger, of the linear mixed effects models of three Viewer2 output variables.

Answer to Reviewer #2

This study is interesting and valuable for investigation of animal behavior and other fields. However, it is not suitable for acceptation in the present form because of the important issues followed:

Answer:

Thank you very much for your valuable feedback and recommending our work for publication. We have revised the manuscript, taking careful account of all your comments.

(1) The authors did not clearly describe the procedure for this investigation sequently in Method, especiallly for the acheivement and processing of the raw data. If possible, why not present a overview of the data processing? In addition, I am also not clear how the Viewer2 work in the present study in this version of manuscript. Is it just a software for data statistic analysis? I think the authors should make some explanation about the Viewer2 in this respect.

Answer: 

The Material and Methods and the results section were reorganized and some information added (See new table of contents below). More details on how the Viewer2 software works are provided in section 2.3.2 lines 232-248. We hope to have clarified now the processing of the raw data. 

2 Material and Methods 7

2.1 Animals and housing 8

2.1.1 Non-experimental condition 8

2.1.2 Pasture 9

2.1.3 Experimental pens 10

2.1.4 Feeding experiment for application of JAM-R 11

2.2 Behaviour recording by observation 11

2.3 Automatic behaviour recording 13

2.3.1 MSR-jaw-movement recording system (JAM-R) 13

2.3.2 Viewer2 behaviour classification software 15

2.4 Comparison of the behaviour recording methods 16

2.4.1 Data management 16

2.4.2 Parameters of agreement 17

2.4.3 Effects of species, condition and recording frequency on agreement parameters 18

2.5 Application of JAM-R in a feeding experiment 20

2.5.1 Description of the feeding experiment 20

2.5.2 Assessment of welfare issues and data quality 21

2.5.3 Estimation of variance of data loggers 22

3 Results 22

3.1 Comparison of the behaviour recording methods 22

3.2 Parameters of agreement 24

3.2.1 Effect of condition: Pasture versus barn (10 Hz) 25

3.2.2 Effect of recording frequency: 10 Hz versus 20 Hz 25

3.3 Application of JAM-R in a feeding experiment 26

3.3.1 Values of feeding and ruminating behaviour 26

3.3.2 Assessment of welfare issues and data quality 26

3.3.3 Variance of data loggers on output variables of the Viewer2 27

(2) The aim of this study seems to validate the performance of the Viewer2 software in classifying the feeding and ruminating behavior in sheep and goat, using the observation by video as the standard for calculation of the efficiency in the system of JAM-R and Viewer2. However, I could not find comparison between the two methods in context. In addition, the data from video observation should also be displayed in supplementary files.

Answer: 

We compared each second of the Viewer2 behaviour classification to the same second of the human behaviour observation classification for (non-)accordance (Lines 264-267, Table 2). From this data the four parameters of agreement were calculated (Section 2.4.2 Lines 287-304) that were further analysed (Sections 2.4.3, Lines 305-335).

For better clarification we reorganized Material and Methods and renamed some headers (see table of contents above). To sum it up, we summarized all information of the behaviour observation in section 2.2 (lines 178ff), all the information on the automatic behvaiour recording and classification system in section 2.3 (lines 205ff) and then described the comparison of these methods in section 2.4., which now reads “Comparison of the behaviour recording methods” (lines 254ff). Additionally, we included the file of comparison of each second of observation to the supplementary data.

(3) Further more, there were some errors in grammar, i.e. line 257-258, as well as formats of typings in the manuscript. 

Answer: 

The manuscript was professionally proof-read. The revised version was again professionally proof-read. 

Answer to Reviewer #3 

This review is for the manuscript titled “Automatic monitoring of feeding and ruminating behaviour of sheep and goats”.

Answer: Thank you very much for your valuable feedback and recommending our work for publication. We have revised the manuscript, taking careful account of all your comments.

Major comments

1) Naturally, goats are browsers and sheep are grazers where their natural grazing behavior can be assessed. Can the authors provide the reason why they used goats on pasture grazing to evaluate feeding behavior study which may not reflect the natural behavior of the animal, especially under grazing condition?

Answer: 

In his classification of ruminant species, Hofmann describes goats as intermediate species, which browse and graze (Hofmann, 1989 ). Grazing is therefore part of their natural feeding behavior. The pasture used for the validation was a natural meadow (see lines 134-136), that allowed browsing to some extent. Under farming conditions, dairy goats are commonly fed on pasture or in the barn. As we did not see relevant differences in the parameters of agreement between species (lines 396-400) we would argue that we were able to assess the feeding behavior of both species equally good for these two conditions. If the system also works for goats under extensive conditions would still have to be shown.

We now mention this restriction in the discussion (Line 507-511): “To cover the full potential of the JAM-R, further aspects need to be investigated. For example, the classification performance in recording periods, where feeding in the barn and on pasture must be distinguished, or the applicability in natural environments, where the predominant feeding activity is browsing rather than grazing.”

2) For the JAM-R recording in feeding pens, a one day recording for 10 and 20 Hz recoding was analyzed (L198-99). Is there any pervious analysis that may suggest the one day recording can generate sufficient data for analyzing the feeding and ruminating behavior in pen situation?

Answer: 

To our knowledge there is no previous analysis available in the literature that would answer how many seconds of recording would be necessary. The observation period in the barn was carefully chosen between 11:00 to 14:00, as this was right after feeding (11:00 am) and was known to include the rumination phase around noon. The sheep and goats were observed to feed on average for 2107 seconds (standard deviation ±1068 s) and ruminate on average for 2678 (±1805) s during these three hours of the observation days in the barn. Furthermore, five animals per species were used to account for variation that might result from individual behavior in feeding. Based on these seconds the parameters accuracy, specificity, sensitivity and precision and their confidence intervals were calculated. As the estimates of the confidence intervals are rather small, (Fig. 3) we would argue that the data set used was at least sufficient for the conditions investigated. For the effect of frequency on the parameters of agreement, 38 observations were used to estimate the effect of two main effects and their interaction in the linear mixed effects model, which leaves sufficient degrees of freedom for the calculation of these effects.

We included the samples size of the data set and subsets in the model formula (lines 313, 322, 329). 

We included this topic in the discussion (Line 479-482): “[…] 10 Hz already provides a good estimate of these behaviours for dairy sheep and goats under the conditions investigated. However, an increase in recording frequency from 10 Hz to 20 Hz slightly improved the classification performance of the Viewer2 in this study.” 

And (line 487-489): “The effect of the recording frequency on the parameters of agreement needs to be tested under additional conditions, such as pasture.”

3) While analyzing the application data, at what recording frequency was the data summarized (min, sec, hr? Line 393) and was that treated as a repeated measure in the model for analyses?

Answer:

Recording frequency was generally 10 Hz (= 10 recordings per second), except for a few recordings in 20 Hz (Lines 342-344). Every single recording is then classified as “feeding”, ‘ruminating’ or ‘other’ by the Viewer 2. The durations for feeding and ruminating per day are calculated by summarizing these classified recordings. The sums are expressed in minutes per recording period (lines 238-239). In case of the feeding experiment the recording period was 24 h (lines 344-345) so the data is expressed as min/d and was calculated to hours/day for a better overview. 

The model in this section was changed to report the variance of the random effects. The included random effects are the data logger, the individual, the feed, the date and the species. The random effect of frequency was not included in this model. This variation is covered in the random effect of data loggers, as only two specific of the 16 data loggers used, were able to record on 20 Hz. For Figure 4 the data was split per hour to visualize the mean feeding and ruminating durations per hour over the day. 

To make this clearer we rephrased the paragraph in M&M. 

It now reads (lines 342-345): “The feeding and ruminating behaviour of each animal was recorded with the JAM-R system on experimental days 4, 5, 9 and 10. The recording frequency was set to 10 Hz, except for 47 cases in which it was set to 20 Hz for the validation described in Section 2.4, which were recorded during the last MR of this experiment. The recording period was set to 24 h from 14:00 to 14:00 the next day”. 

And (line 372-377): “In the next step, the order of magnitude of the variance attributed to the individual data loggers (n = 18) was estimated. Three output variables of Viewer2, the feeding duration in hours per day, rumination duration in hours per day and mastications per cud (MpC), were analysed each in a linear mixed effects model (lme4 package, REF) to estimate the variance of the random effect data logger. Other random effects for comparison included in the model were individual, species, the feed (HH, GH, MG) and the date.”

Minor comments

Abstract

1) Line 39-41: review as “… system with the Viewer2 software provided a reliable technology for automatic recording of feeding and ruminating behavior of sheep and goats on pasture and in the barn”.

Answer: 

done.

Introduction

2) The introduction section is a bit long and requires revision to make it concise and also increase its flow.

Answer: 

We shortened the section related to the use of behaviour parameters (Lines 44-51).

3) Line 45: revise as “… monitoring feeding and ruminating behavior of ruminants is …”

Answer: 

Changed to (lines 43-44): “In ruminants, monitoring feeding behaviours is useful for assessing their health and welfare.”

4) R3: Lines 61-64: please revise this sentence is not clear

Answer: 

As this section of the introduction was shortened, this sentence was removed.

5) R3: L65; … feeding and rumination ….

Answer: 

To keep the wording concise, we use “ruminating” throughout the entire manuscript.

6) R3: L69-73: this seems to be a single sentence and can be revised as “…(15) including; 1) …; 2)…..; and 3)…”

Answer: 

We decided to to avoid sentences that span over more than three lines. We rephrased the section. It now reads (lines 57-63): “Devices that record actual jaw movements with pressure-sensitive sensors have been designed repeatedly because this approach has a number of advantages (9). First, it measures behaviour directly at the individual level, so the data can always be linked to the individual, independent of the animal’s location (barn or pasture). Second, the available sensors are small enough to fit into common head collars, so animals can easily adapt to the system. Third, the measurement of jaw movements is at the exact location where the behaviour of interest originates.”

7) L74-76: revise as “… feeding and ruminating behaviours as the duration and frequency of mastication differs.”$

Answer: 

We rephrased the sentence. In now reads (lines 64-66): “This enables a distinction between feeding and ruminating, as the duration and frequency of mastication differ between these two behaviours.”

8) L79: revise as “… (18). The system was tested and validated for dairy cows (19).”

Answer: 

We rephrased the sentence. It now reads (lines 69-70): “The JAM-R technology has previously been tested and validated for dairy cows.”

Material & Methods

9) L108: revise “To validate the software Viewer2, feeding and ….”

Answer: 

We rephrased the sentence. It now reads (lines 98-100): “To validate the JAM-R, the behaviour classifications of Viewer2 software (described in Section 2.3.2) were compared to the gold standard method of behaviour observation (described in Section 2.2).”

10) L116: delete respectively

Answer: 

The paragraph was revised. It now reads (Lines 102-105): “The agreement between the classifications of Viewer2 and the behaviour observations was evaluated. Additionally, JAM-R was applied in a feeding experiment with multiple 24 h recordings and assessed for technical and welfare issues.”

11) L121-22: revise this sentences. Is this means 2 goats (from pasture) for observation, 5 goats (including the two) for video, and 5 sheep for video monitoring used? Also, clarify if the animals used for monitoring behavior on pasture and pen settings were the same.

Answer: 

We rephrased the sentence. It now reads (lines 151-154): “Video recordings of five sheep and five goats ([…]) were collected [...]. The animals were not the same as those used on pasture, apart from two goats that were used in both situations.”

12) L114-131: include the average body weight (±SD) of the experimental sheep and goats

Answer: 

The information was added (lines 116-117): “The mean weight of goats was 67.7 (SD ± 7.5) kg and of sheep was 79.1 (± 7.7).”

13) L140: revise “… pend had three drinking water sources and one mineral supply site.”

Answer: 

done.

14) L156: Is there a reason why the sheep were moved to the same plots that is already used by the goats earlier?

Answer: 

This was necessary to enable water supply. However, sheep received additionally fresh pasture for habituation and during the experiment. (Line 147)

15) L164-65: just out of curiosity, what is the value of having a round table in the pens?

Answer: 

The elevated tables and platforms are provided for goats to structure the pens to give possibility for the goats to avoid each other and give climbing possibilities. For further information see Aschwanden et al. 2009a and Aschwanden et al. 2009b . 

16) L165: revise as “ Experimental animals received a MR containing 55% …… ad libitum.”

Answer: 

Done.

17) L167: Naturally, goats are browsers and sheep are grazers where their natural grazing behavior can be assessed. Can the authors provide the reason why they used goats on pasture grazing to evaluate feeding behavior study which may not reflect the actual natural behavior of the animal?

Answer: 

See Answer to the Major Comment 1).

18) 

a) L172-73: revise as “… on the weather condition, ….”.

Answer: 

Done.

b) During higher temperature, animals tend to be less active and gather under a shade/ hut and this is one of the occasions that rumination takes place. So, during grazing in the field, is the observation in the morning specifically focus on feeding and the one in the afternoon focused on rumination?

Answer: 

The animals grazed and ruminated in several phases throughout the whole day. We observed in observation periods of 1.5 to 3.5 hours. Most of the time both behaviors could be observed in such an observation period. 

19) L177: which previous experiment? Include the source. This may also answer the question if the two days of recording can generate sufficient data for behavioral study. As indicated in L198, basically it is a one day data as different recording frequency (10 vs 20 Hz) was used on separate days.

Answer: 

The data for validation in the barn was collected during testing the third mixed ration in the feeding experiment described in section “2.5 Assessment of application of JAM-R in a feeding experiment”. During enrolling the first two mixed rations we assessed during which hours the main phases of feeding and rumination will occur. From this information, we chose the period 11:00 to 14:00.

To make this clear, we added the information. 

Lines 151-153: “Video recordings of five sheep and five goats […] were collected during the feeding experiment which will be further explained in Section 2.1.4.”

Lines 176-177: “During the experimental phase of (the last MR) MG, videos for behaviour observations (Section 2.2) were recorded.”

20) L205: different font

Answer: 

Corrected.

21) L218: revise as “ …file (one per animal per day) ..”

Answer: 

It now reads (lines 229): “the single MSR files must be uploaded to Viewer2.”

22) L226: Based on L144 L160, and L301, all the grazing and pen feeding experiments were conducted in 2020 however, in the excel spreadsheet provided here (under application tub in the file) the date stamp indicated data collected in 2019.

Answer: 

The experimental time was incorrectly stated. The experiment for the application was conducted from October 2019 to April 2020. This was corrected. (Line 165)

23) L300: Is this a separate experiment from the one stated in L159-66 for pen feeding that used just five animals, March to April 2020?

Answer: 

This was actually the same experiment. We clarified this. Please see answer to point (19).

24) L306-7: Explain the reason for providing different MRs? Also, revise – a 14 days adaptation period was mentioned on L314 – so perhaps a specific MR was provided for a total of 24 days (14 adaptation and 10 experimental).

Answer: 

The objective of the feeding experiment required that three different MRs common in small ruminants under intensive housing conditions were studied. It additionally allowed us to collect data for the validation of the JAM-R-system during feeding of the third MR. 

The sections of M&M were restructured according to the comments of Reviewer 1. We hope to have clarified these issues now. 

Information on the feeding experiment is now in Section 2.5. Every ration was fed for 10 days with a preceding habituation phase of 14 days. (Lines 174-176)

25) L324-5: what was the recording frequency considered for JAM-R (min, sec, hr? L393) and was that tread as a repeated measure in the model for analyses?

Answer: 

See answer to Major comment 3)

26) L326: please include the quality analysis results for the grazed pasture as well

Answer: 

According to the modifications of this section suggested by Reviewer 1, information on the quality analysis of the MRs was moved to the supplementary section. There are no specific analysis for the pasture available. Pasture quality is described in lines 134-136. For the validation of the system, we investigated if the parameters of agreement differed between pasture and the barn (which was based only on the third MR) (Line 162). Potential effects of feed composition on the parameters of agreement would have to be tested in a follow-up experiment based on specific hypotheses. This was not the scope of our study. Concerning the results of the application in the feeding experiment, an influence of the different qualities of the MRs on the outcome variables (feeding duration, rumination duration and bites per cud) can be expected and the effect of feed was included as random effect. (Lines 380, Results Tab. 5). 

No changes made.

Results

27) L341: The system seems a poor predictor for drinking (Table 4) especially on pasture. Include descriptions on drinking behavior as well.

Answer: 

The system does not predict drinking behaviour. According to the specific pressure patterns “feeding” and “ruminating” is classified. All seconds non-classified as feeding or ruminating are considered as “no / other behaviour”. The correct classification for drinking behaviour would be “no/ other behaviour”. In our observation we included drinking to see whether it would be correctly classified as “no/ other behvaiour”.

The explanation of the Viewer2 software and its classification were reformulated and expanded (see Section 2.3.2). We also rephrased the information in the results section (lines 387-389): “The seconds at which no behaviour or other behaviour was observed and the software should “ignore” were correctly classified as “no or other behaviour” in 89.5% of seconds (Table 4).”

28) L347: revise as “… were not correctly classified …”. As the system depends on change in pressure (with no visual aid to specifically identify and label activities), the observed

Answer: 

See answer to comment 27. 

29) L373 & 384: include the details of the results reported under these sections in a table form, perhaps expanded on Table 4 or added as supplementary table.

Answer: 

The statistical results were added as Supplementary Table 3.

30) L390: revise as “On average (±SD), sheep fed for 5.1….. MpC across the three MRs. Conversely, goats fed ….”

Answer: 

Done.

31) L403-4: On L303-4, stated that data was collected from 26 sheep and goats but only results from 24 is reported here.

Answer: 

There are 24 sheep and 24 goats as experimental unit of the experiment, but two per species were replaced during the experiment. 

The information was added in the methods section (lines 168-169).

32) L512: References

Answer: 

Done.

Tables

33) Table 5: Include the unit for dry matter intake

Answer: 

This table was excluded in the new version of the manuscript. The DM intake information with its unit was included in the text.

It now reads (lines 432-434): “On average (± SD), sheep consumed 1.34 (± 0.13) kg DM in 4.9 (± 1.0) h/day, ruminated for 7.2 (± 1.4) h/day and had 70.43 (± 10.7) MpC across the three MRs.”

34) Supplementary 1 (word file)

The reported size of area and the numbers placed on the demarcated landscape doesn`t match. For example, for observation goats site the number on the map is 100 m by 119.55 m but the reported area 503.58 m2. The same for the sheep sites as well.

Answer: 

The circumference of the plot “observation goats” is 119.55 m as it is stated in the box below and the 100 m point was marked. We removed this mark for 100 m to avoid this confusion.

35) Supplementary (excel file)

As stated above, the data collection year needs to be checked (2019 vs 2020). The headings for individual columns within each tabs needs to be clearly presented/stated. For example, in the ‘application’ tab ‘starttime’ in column A is indicated as 140000 and in column J and K separate date and time reported.

Answer: 

The experimental phase of the application was stated wrong. It is corrected and lasted from October 2019 to April 2020 (line 165). The supplementary data table was reorganized: The column ‘starttime’ was deleted. It was from the original file, which is not used in the analysis.

---

## [Decision Letter · Decision Letter 1]

21 Mar 2023

PONE-D-22-30895R1Validation of automatic monitoring of feeding behaviours in sheep and goatsPLOS ONE

Dear Dr. Berthel,

Thank you for submitting your manuscript to PLOS ONE. After careful consideration, we feel that it has merit but does not fully meet PLOS ONE’s publication criteria as it currently stands. Therefore, we invite you to submit a revised version of the manuscript that addresses the points raised during the review process. One of the reviewer require major revision before the publication of the manuscript. This manuscript will not be published unless all the comments that has been raised by reviewer addressed properly. Please submit your revised manuscript by May 05 2023 11:59PM. If you will need more time than this to complete your revisions, please reply to this message or contact the journal office at plosone@plos.org. Please include the following items when submitting your revised manuscript:A rebuttal letter that responds to each point raised by the academic editor and reviewer(s). You should upload this letter as a separate file labeled 'Response to Reviewers'.A marked-up copy of your manuscript that highlights changes made to the original version. You should upload this as a separate file labeled 'Revised Manuscript with Track Changes'.An unmarked version of your revised paper without tracked changes. You should upload this as a separate file labeled 'Manuscript'.

We look forward to receiving your revised manuscript.

Kind regards,

Aziz ur Rahman Muhammad

Academic Editor

PLOS ONE

Additional Editor Comments:

Dear Authors,

One of the reviewer again requested for major revision. Therefore, i would like to decide that this manuscript should undergo major revision unless all the comments that has been raised by reviewer addressed properly

Reviewers' comments:

Reviewer's Responses to Questions

**Comments to the Author**

1. If the authors have adequately addressed your comments raised in a previous round of review and you feel that this manuscript is now acceptable for publication, you may indicate that here to bypass the “Comments to the Author” section, enter your conflict of interest statement in the “Confidential to Editor” section, and submit your "Accept" recommendation.

Reviewer #1: (No Response)

2. Is the manuscript technically sound, and do the data support the conclusions?

Reviewer #1: Partly

3. Has the statistical analysis been performed appropriately and rigorously? 

Reviewer #1: Yes

4. Have the authors made all data underlying the findings in their manuscript fully available?

Reviewer #1: Yes

5. Is the manuscript presented in an intelligible fashion and written in standard English?

Reviewer #1: Yes

6. Review Comments to the Author

Reviewer #1: General comments: First off I would like commend the authors for the great effort of revising this manuscript. I find that the new structure looks better and the manuscript has improved with the new additions and clarifications.

That said, I do still have some additional concerns with the manuscript in its current state, in particular the data cleaning section which could have implication on both the current results and its discussion. Thank you for the effort so far.

Please see line by line below.

Line 55-57. I would consider writing out examples of these mechanisms.

Line 98-100. I would consider clarifying briefly what the gold standard is in this paragraph. I would also make sure the reader understands what the gold standard is in section 2.2. Currently you describe both manual live observation and manual observation from continuous recordings, but you don’t define the gold standard. Please clarify.

Line 152-153. Why did two goats re-enroll in the project? Were they simply filler animals to keep the group size intact?

Line 179-180. I think it would be more clear if you simply wrote out 6h instead of 21,600 seconds. Were the videos 3h each on the same time of day across two days, or on two occasions on the same day? How were these periods selected, by choosing parts of the day when the sheep and goats where most likely to display the behaviors in the ethogram? Was it during the same time frame as written later on lines 190-193?

Line 196: Is 2h of non-continuous data per animal enough to properly validate the logger?

Line 189-203. If I understand correctly the animals on pasture were not recorded using a gold standard, as a 10 minute intervals were used, compared to the pens which had continuous recordings from video? If continuous interval focal animal sampling is the gold standard for goats and sheep on pasture, please be very clear in the manuscript and provide a reference.

Line 218: “The data loggers HAD to be programmed…”

Line 218-221: Did you ever consider looking at higher frequencies as well? You mentioned earlier that the data logger could be programmed up to 50 hz.

Line 221-223. While re-reading the manuscript, I got curious to why the activity was omitted from the manuscript?

Line 223-225. So this is quite the limitation of this particular logger if you wanted to record continuous data over a longer period of time without disturbing the animals. Would there be enough utility for this logger when you factor in the task of changing these loggers daily? In line 226, you mention other loggers, do you mean from the same company/brand with longer battery life, or other loggers from other companies? If the latter, you might want to be careful in how this limitation is portrayed as you are essentially promoting other companies loggers with longer battery life. Especially since the company co-authoring the manuscript.

Line 255-258. So if I understand correctly, the data from the pasture on 10 hz was compared to a gold standard, and for animals in the pen, both a 10 hz and a 20 hz data set was compared to a gold standard.

Line 264-266. Please be clear with defining this ‘gold standard’ is.

Line 306-310. Consider breaking this sentence up for clarity. Also consider replacing the word “checked” for ‘controlled’ depending on the new sentences.

Line 363-364: By excluding these errors, are you not excluding data that would tell you the performance of these loggers, i.e. selection bias?

Line 364-370: You state that some of the feeding and ruminating durations of less than 3.1h/d were implausible and that the cut-off values are based on minimal durations for feeding in goats and small ruminants. Under what conditions? The same? The Lu reference refers to an indoor feeding trial, and Penning recorded pasture based feeding durations of up to 11.4h for goats and 12.6h for sheep. Way over the exclusion threshold made by your outlier calculation. Then you state that implausible high durations for feeding and ruminating were identified by outliers, and in the next sentence you state that no outliers were found. This would be the case if ‘implausible values’ were omitted from the data set as previously stated. You might want to revise/control how this data set was cleaned and using what rationale. Ergo, perhaps the data omitted was not implausible after all? I would consider looking at this data again to make sure the calculated accuracy, specificity and precision are valid and derived from a correctly cleaned data set.

Line 369. Square root.

Line 442-461. Not

Line 458-461. See previous comment for lines 364-370. I think the majority of this paragraph can be condensed and merged in to section 2.5.2 and the few welfare aspects regarding the logger placement can be discussed in the discussion.

Table 2. Make sure to clarify what type of video recording that was used. Continuous?

7. PLOS authors have the option to publish the peer review history of their article (what does this mean?). If published, this will include your full peer review and any attached files.

Reviewer #1: No

---

## [Author Response · Author response to Decision Letter 1]

21 Apr 2023

General Answer:

Thank you very much for your valuable feedback and recommending our work for publication. 

We have revised the manuscript, taking careful account of your comments.

1) Line 55-57. I would consider writing out examples of these mechanisms.

Answer:

We added examples (line 58 – 60): “In the past, automatic monitoring systems were developed based on a variety of mechanisms, like head position with accelerometers or jaw movements with acoustic or pressure sensors.”

2) Line 98-100. I would consider clarifying briefly what the gold standard is in this paragraph. I would also make sure the reader understands what the gold standard is in section 2.2. Currently you describe both manual live observation and manual observation from continuous recordings, but you don’t define the gold standard. Please clarify.

Answer:

We added the definition of the “gold standard” in the introduction (lines 53 – 58): “Despite the rise of technological solutions, behaviour observation by human observers is still commonly used to record animal behaviour and can be considered the “gold standard”. However, it requires training and is very time consuming, making it challenging in research and very demanding on farms. Automated recording methods can substitute the human observer provided they are equally good at detecting feeding behaviour as humans.”

We rephrased (line 102 – 104): “To validate the JAM-R, the behaviour classifications of Viewer2 software (described in Section 2.3.2) were compared to the gold standard of behaviour recording, which is the observation by a human (detailed methods used is described in Section 2.2).”

We added in section 2.2 (lines 185 – 187): “During the behaviour observations which is considered the gold standard, all animals were equipped with the reference behaviour recording method - the JAM-R.”

3) Line 152-153. Why did two goats re-enroll in the project? Were they simply filler animals to keep the group size intact?

Answer: 

There is no particular reason why the two goats were re-enrolled in the project; it was a coincidence that they were included in both situations. We had videos from most of the animals. Unfortunately, many videos had a bad quality in terms of lighting or because the head of the animal was hidden behind the other animal or barn structures for some time. The animals chosen were simply those, where we had videos of two required days (once 10 Hz and once 20 Hz) that could be coded (= good lighting and animals head in sight the entire time).

We rephrased the information of the animals chosen for the barn validation (lines 154 – 159):“During the feeding experiment, which will be further explained in Section 2.1.4., videos were recorded. Videos of five sheep and five goats (LA = 4, EF = 1, CC = 2, SA = 3) were chosen based on the availability of sufficient video image quality (e.g. lighting, animal in sight) for continuous behaviour observation. This included two goats that were also used in the pasture situation and three other goats and five additional sheep.”

4) Line 179-180. I think it would be more clear if you simply wrote out 6h instead of 21,600 seconds. Were the videos 3h each on the same time of day across two days, or on two occasions on the same day? How were these periods selected, by choosing parts of the day when the sheep and goats where most likely to display the behaviors in the ethogram? Was it during the same time frame as written later on lines 190-193?

Answer:

We indeed checked each single of 21’600 seconds. They were coded from videos of two days of 3 hours of two different animals. We moved this section below the explanation of the video observation procedure, where it is explained which periods were chosen and why and rephrasedfor clarification.

It now reads (lines 206 – 210): “The behaviour observations on pasture (direct) and in the barn (by video) were conducted by one trained observer (A.D.). Two of the videos were coded again by another trained observer (R.B.) and each second (21’600 s) was assessed for agreement. R.B. reached an agreement of 97.6–98.2% for seconds observed as feeding by A.D. and 98.5–99.9% for seconds of rumination of A.D.”

5) Line 196: Is 2h of non-continuous data per animal enough to properly validate the logger?

Answer:

The data was recorded continuously. Every 10 min we switched to another animal adding up to 22.8 hours (sheep) and 22.7 hours (goat) of continuous observation in total. The validation of the software Viewer2 is therefore based on the classified seconds of 45,5 hours (22.8 + 22.7 h) of continuous recording (see point 6). In Section 2.5, we focus on the variance between the loggers, which is based on 401 24h recordings during the feeding experiment, summing up to 9’624 hours.

No changes made.

6) Line 189-203. If I understand correctly the animals on pasture were not recorded using a gold standard, as a 10 minute intervals were used, compared to the pens which had continuous recordings from video? If continuous interval focal animal sampling is the gold standard for goats and sheep on pasture, please be very clear in the manuscript and provide a reference.

Answer:

Apparently we were not clear enough. We rephrased that sentence and included, that the 10 min intervals of pasture observation were indeed also continuous 10 min behaviour observations. Per animal data of 3 hours of video observation and 2.3 hours of direct observation were collected (Line 203 and Line 109).

It now reads (line 195-197): “. Using focal animal sampling, the observer continuously observed the animals by switching between animals in 10 min intervals or when the head of the focal animal moved out of sight.”

7) Line 218: “The data loggers HAD to be programmed…”

Answer: done.

8) Line 218-221: Did you ever consider looking at higher frequencies as well? You mentioned earlier that the data logger could be programmed up to 50 hz.

Answer: 

As we only had access to an older generation of MSR loggers for this project, our loggers were not able to record on 50 Hz. As discussed in lines 498 – 500, we suggest the comparison of higher frequencies for future research (“For a more detailed analysis of ruminating and feeding behaviour, and depending on the research question, a recording frequency of 20 Hz (or even higher) could therefore be advisable.”)

No changes made.

9) Line 221-223. While re-reading the manuscript, I got curious to why the activity was omitted from the manuscript?

Answer:

This data was not omitted, we did not collect it. Measuring activity was not within the scope of this study and we needed the available memory capacity for the 20 hz recordings. 

No changes made.

10) Line 223-225. So this is quite the limitation of this particular logger if you wanted to record continuous data over a longer period of time without disturbing the animals. Would there be enough utility for this logger when you factor in the task of changing these loggers daily? In line 226, you mention other loggers, do you mean from the same company/brand with longer battery life, or other loggers from other companies? If the latter, you might want to be careful in how this limitation is portrayed as you are essentially promoting other companies loggers with longer battery life. Especially since the company co-authoring the manuscript.

Answer: 

For sustainability reasons, we use electronic equipment as long as it is functioning. Therefore, the loggers used in this experiment were comparably old (purchased 2009 - 2010, see line 449). Newer loggers are available from MSR electronics GmbH. This information was included.

It now reads (lines 230 – 234): “The data loggers (in the version available for this project) had to be removed from the animals to save the recorded data (as an MSR file) and reprogrammed for the next recording period once per day due to the data storage capacities of the loggers being a maximum of 29 hours. However, newer versions of the data loggers are available with larger data storage capacities for longer continuous recording periods from MSR electronics GmbH.”

11) Line 255-258. So if I understand correctly, the data from the pasture on 10 hz was compared to a gold standard, and for animals in the pen, both a 10 hz and a 20 hz data set was compared to a gold standard.

Answers:

This is correct. The automatically recorded data classified into behavior by the Viewer2 was compared to the behaviour recorded by the human observer.

We added an introductory sentence in this section (lines2– 267): “The behaviour recorded by the human observer (directly on pasture or from video in the barn, see Ethogramm in Tab. 1) was compared to the behaviour recorded at the same time with the data loggers and classified by the Viewer2. The data loggers were set to a recording frequency of 10 Hz when the animals were on pasture. In the barn, the recording frequency was alternated between 10 and 20 Hz, so 

every animal had one day of recording at 10 Hz and one day of recording at 20 Hz.”

12) Line 264-266. Please be clear with defining this ‘gold standard’ is.

Answer:

We rephrased (lines 275 – 276): “This resulted in numbers of correctly and incorrectly classified seconds, taking the behaviour observed by the human as the true value (“gold standard”).”

Please also see our answer to point 2).

13) Line 306-310. Consider breaking this sentence up for clarity. Also consider replacing the word “checked” for ‘controlled’ depending on the new sentences.

Answer:

We changed to (lines 315-318): “With the complete dataset, in linear mixed effects models (lmer of lme4 package, (24); one per parameter), as outcome variables, each parameter of agreement was estimated in dependence of the fixed effects species (sheep and goat) and behaviour (feeding and ruminating). The models also considered the repeated measurements of the individual, the effect of the condition and the recording frequency by random effects.”

14) Line 363-364: By excluding these errors, are you not excluding data that would tell you the performance of these loggers, i.e. selection bias?

Answer:

As reported in the first part of the paper (the validation of the Viewer2), the software does not classify behaviour 100 % correctly. Therefore any dataset generated with the JAM-R, needs to be checked for implausible values to improve data quality. In this section of the paper (application in a feeding experiment), our purpose was to provide the reader (that potentially wants to use the system to record feeding behaviour) with information how much data loss, technical failure and variation in the data caused by a set of these data loggers can be expected; or better: need to be considered. From our point of view to report on the variation in the data caused by the loggers, the information is more useful after identifying and removing implausible values. The procedure must of course be explained and the number of removed data has to be documented (line xx-xx).

We added information to explain better the aim of the application in a feeding experiment (lines 346-350): “To assess issues of the applicability of the JAM-R, it was tested in a feeding experiment. Issues for animal welfare due to wearing the halters were recorded. Data loss events were reported which were caused by technical failure, animal welfare issues or human error. Raw data of the feeding and rumination durations were inspected for plausibility. The 

variation in the data that was caused by this set of data loggers was assessed.”

15)

15.1) Line 364-370: You state that some of the feeding and ruminating durations of less than 3.1h/d were implausible and that the cut-off values are based on minimal durations for feeding in goats and small ruminants. Under what conditions? The same? The Lu reference refers to an indoor feeding trial, and Penning recorded pasture based feeding durations of up to 11.4h for goats and 12.6h for sheep. Way over the exclusion threshold made by your outlier calculation. Then you state that implausible high durations for feeding and ruminating were identified by outliers, and in the next sentence you state that no outliers were found. This would be the case if ‘implausible values’ were omitted from the data set as previously stated. 

15.2) You might want to revise/control how this data set was cleaned and using what rationale. Ergo, perhaps the data omitted was not implausible after all? I would consider looking at this data again to make sure the calculated accuracy, specificity and precision are valid and derived from a correctly cleaned data set.

Answer:

1) We aimed at choosing cut- off values for implausible values as conservatively as possible and specifically tailored to our experimental approach. For implausible low values, we took the minimal duration that we could find for small in the literature. Reports on maximum durations are rare, in a broad range and do not match our experimental conditions (indoor vs pasture, individual vs group housing, different feeds). For this reason, we used the mathematical definition of outliers for implausible high values. For rumination, we did not have such outliers in the data. 

2) For the calculations for accuracy, sensitivity, precision and specificity we compared the classifications of single seconds and did not calculate durations. Therefore, we had no implausible values and did not exclude any data in that section of the paper.We clarified that the used cut-off values are not a general approach, but based on our experimental approach. It now reads (lines 373-382): “Thereafter, the Viewer2 output data files were analysed for implausible values that would compromise data quality. It was assumed that data files of extreme values for feeding and/or ruminating duration would indicate incorrect classifications in these specific files. For the purpose of the experimental approach of this feeding experiment, data files with feeding and ruminating durations of less than 3.1 h/d were deemed implausible. These cut-off values were set from the minimal durations for feeding in goats reported by Lu (26) and for ruminating in small ruminants reported by Penning et al. (27). 

Due to the lack of suitable literature, implausible high durations for feeding and ruminating were identified by outliers (+/-1.58 interquartile range/square root (n), (28)), which excluded feeding durations exceeding 8.1 h/d and rumination durations higher than 10.7 h/d.”

For clarification, we now state all number of outliers for feeding and rumination duration in the results section (line 475-478). “From the remaining 510 MSR files that were analysed by Viewer2, 109 (21.4%) output data files were identified to have implausible high (feeding: 23; ruminating: 0) or low (feeding: 25; ruminating: 81) durations of feeding and ruminating, resulting in a final dataset of 401 data files. This represents 71.4% of the executed 562 MSR files.”

16) Line 369. Square root.

Answer: done.

17) Line 442-461. Not

Answer:

We assume you intended to write more. Please clarify.

18) Line 458-461. See previous comment for lines 364-370. I think the majority of this paragraph can be condensed and merged in to section 2.5.2 and the few welfare aspects regarding the logger placement can be discussed in the discussion.

Answer:

We would argue that these results are part of the technical evaluation next to the logger variation and prefer to keep it here. See also our answer to comment 14).

We rephrased parts of the associated methods section and this results section and the respective headings (2.5.2 and 3.3.2) to be clearer about the purpose of these sections.

The section headings were change to (line 361 + 454): “Assessment of welfare issues, failure rate and data plausibility”

It now reads (lines 369– 382): “It was noted daily, if a recording was not executed due to human error or animal welfare issues. Furthermore, all technical failures during the feeding experiment from all three MRs were recorded. These were identified by the number of MSR files that were not readable by Viewer2. Thereafter, the Viewer2 output data files were analysed for plausibility that would compromise data quality. It was assumed that data files of extreme values for feeding and/or ruminating duration would indicate incorrect classifications in these specific files. For the purpose of the experimental approach of this feeding experiment, data files with feeding and ruminating durations of less than 3.1 h/d were deemed implausible. These cut-off values were set from the minimal durations (known to the authors) for feeding in goats reported by Lu (26) and for ruminating in small ruminants reported by Penning et al. (27). Due to the lack of suitable literature, implausible high durations for feeding and ruminating were identified by outliers (+/- 1.58 interquartile range/square root (n), (28)), which excluded feeding and ruminating durations exceeding 8.1 h/d and 10.7 h/d, respectively.”

This paragraph was moved from Materials and Methods to the results section (now lines 465-469): “From the 576 scheduled MSR files, in 14 cases (2.4%), the MSR file was missing due to human error (data loggers were unintentionally not programmed) or animal health (one animal could not wear the halter for the entire experiment due to an abscess on the jaw, which ruptured when trying on the halter the first time and then needed to heal), resulting in the analysis of 562 24-hour MSR files.”

19) Table 2. Make sure to clarify what type of video recording that was used. Continuous?

Answer:

We clarified that both conditions of behaviour observation were continuous observations (see point 6) and added this information in the description of table 2 (line 294)

---

## [Decision Letter · Decision Letter 2]

5 May 2023

Validation of automatic monitoring of feeding behaviours in sheep and goats

PONE-D-22-30895R2

Dear Dr. Berthel,

We’re pleased to inform you that your manuscript has been judged scientifically suitable for publication and will be formally accepted for publication once it meets all outstanding technical requirements.

Kind regards,

Aziz ur Rahman Muhammad

Academic Editor

PLOS ONE

Additional Editor Comments (optional):

Dear Authors,

Thanks for incorporating all the comments raised by reviewers. Good work and good luck

Regards

Reviewers' comments:

Reviewer's Responses to Questions

**Comments to the Author**

1. If the authors have adequately addressed your comments raised in a previous round of review and you feel that this manuscript is now acceptable for publication, you may indicate that here to bypass the “Comments to the Author” section, enter your conflict of interest statement in the “Confidential to Editor” section, and submit your "Accept" recommendation.

Reviewer #1: All comments have been addressed

2. Is the manuscript technically sound, and do the data support the conclusions?

Reviewer #1: Yes

3. Has the statistical analysis been performed appropriately and rigorously? 

Reviewer #1: Yes

4. Have the authors made all data underlying the findings in their manuscript fully available?

Reviewer #1: Yes

5. Is the manuscript presented in an intelligible fashion and written in standard English?

Reviewer #1: Yes

6. Review Comments to the Author

Reviewer #1: The authors have answered and clarified all my questions in this round of the reviews. I appreciate the authors' time spent on revising the manuscript. Good work!

7. PLOS authors have the option to publish the peer review history of their article (what does this mean?). If published, this will include your full peer review and any attached files.

Reviewer #1: No

---

## [Editor Report · Acceptance letter]

9 May 2023

PONE-D-22-30895R2 

Validation of automatic monitoring of feeding behaviours in sheep and goats 

Dear Dr. Berthel:

I'm pleased to inform you that your manuscript has been deemed suitable for publication in PLOS ONE. Congratulations! Your manuscript is now with our production department. 

Kind regards, 

on behalf of

Dr. Aziz ur Rahman Muhammad 

Academic Editor

PLOS ONE